# Physics-based forecasting of man-made earthquake hazards in Oklahoma and Kansas

Cornelius Langenbruch [1], Matthew Weingarten[1,2] & Mark D. Zoback [1]

Reinjection of saltwater, co-produced with oil, triggered thousands of widely felt and several damaging earthquakes in Oklahoma and Kansas. The future seismic hazard remains uncertain. Here, we present a new methodology to forecast the probability of damaging induced earthquakes in space and time. In our hybrid physical–statistical model, seismicity is driven by the rate of injection-induced pressure increases at any given location and spatial variations in the number and stress state of preexisting basement faults affected by the pressure increase. If current injection practices continue, earthquake hazards are expected to decrease slowly. Approximately 190, 130 and 100 widely felt $M \geq 3$ earthquakes are anticipated in 2018, 2019 and 2020, respectively, with corresponding probabilities of potentially damaging $M \geq 5$ earthquakes of 32, 24 and 19%. We identify areas where produced-water injection is more likely to cause seismicity. Our methodology can be used to evaluate future injection scenarios intended to mitigate seismic hazards.

[1] Department of Geophysics, Stanford University, Stanford, CA 94305, USA. [2]Present address: Department of Geological Sciences, San Diego State University, San Diego, CA 92182, USA. Correspondence and requests for materials should be addressed to C.L. (email: langenbr@stanford.edu)

Probabilistic seismic hazard analyses (PSHA) has been used to develop earthquake hazard maps in intraplate areas for several decades[1–4]. Traditionally, PSHA is used to develop maps of the probability of strong ground shaking over relatively long periods of time. It has been widely used by governments and industry in applications such as assessing the safety of nuclear power plants, developing building code requirements and determining earthquake insurance rates. The methodology assumes long-term stationarity of seismicity rates resulting from large-scale geologic processes. Application of these methods to man-made earthquakes, especially those induced by underground injection of fluids, is inherently problematic because injection rates (and thus induced earthquake rates) can vary markedly in space and time[5–7]. For example, over the past 6 years, north-central Oklahoma and southernmost Kansas experienced thousands of widely felt earthquakes ($M \geq 3$) resulting from injection of very large amounts of saltwater that was co-produced with oil[6–9]. The seismicity triggered by the fluid injection corresponds to >2000 years of natural tectonic activity[7] and occurred due to the decrease of frictional resistance to slip on tectonically loaded, preexisting faults[10].

Seismic activity in north-central Oklahoma and southernmost Kansas peaked in 2015, when 943 widely felt $M \geq 3$ (31 $M \geq 4$) earthquakes occurred[11] in response to the dramatic increase in produced saltwater injection rates into the basal sedimentary Arbuckle Group[6,7] (Fig. 1a, b). Driven by market forces and mandated reductions[12], injection rates started to decrease rapidly in mid-2015. Compared to peak injection rates, a reduction of about 50% was reported through March 2018 (Fig. 1b). The earthquake rate responded to decreased saltwater injection rates and reduced by about 80%[7,13] (Fig. 1b). While the overall earthquake rate has decreased markedly over the past 2 years, the seismic hazard remains high[7,13,14]. In 2017, 294 $M \geq 3$ (6 $M \geq 4$) earthquakes were recorded[11]. To date, four induced earthquakes in the study area exceeded $M = 5$, including the September 2016 Pawnee $M = 5.8$ earthquake—the largest in instrumented history in Oklahoma and Kansas.

While the utilization of PSHA has been questioned, in general[15], the main problem when being applied to injection-induced seismicity is that changes of the driving force, and variations of injection rates in space and time, are not considered. More recently, a 1-year seismic hazard model has been developed for Oklahoma and Kansas using last year's seismicity rates to predict the seismic hazard[14]. While a step in the right direction, we argue here that induced seismic hazard should be forecasted based on a physical understanding of induced earthquakes.

In the sections below, we outline a physics-based model for estimating spatial and temporal variations of seismic hazards in Oklahoma and Kansas. The model has two principal components. First, spatial and temporal variations of injection-induced pressure changes are evaluated utilizing the reported injection rates and a regional hydrogeologic model. Second, spatial variations of the seismogenic state are considered, a proxy for number and stress state of preexisting basement faults that are affected by the pressure increase. In this regard, the study reported here departs significantly from our previous analysis of seismicity rates in Oklahoma[7] using a seismogenic index model[16]. First, we integrate a regional hydrologic model into the analysis and second, we assess spatial variability of the seismic hazard. After combining spatial and temporal variations of injection-induced pressure changes and spatial variations of the seismogenic state in a hybrid physics-based statistical model, we create probabilistic seismic hazard maps

and an estimate of the earthquake probability throughout the entire region through 2020.

## Results

**Injection-induced pressure increase at seismogenic depth.** To compute injection-induced pore pressure changes at depth, we modelled 809 Arbuckle wells injecting at about 2.1-km depth from Jan 2000 through March 2018 (see Supplementary Fig. 1). The Arbuckle Group is directly overlying the crystalline basement and consists of a pervasively fractured, dolomitic carbonate aquifer with hydraulic continuity over tens to hundreds of kilometres[17]. Reported permeability ranges from core, outcrop and field-scale measurements in Oklahoma and Kansas show permeability as low as $10^{-14}$ m$^2$ and as high as $3 \times 10^{-12}$ m$^2$[18–22]. As explained in the Methods section, we represent the major lithologic units present across Oklahoma and Kansas with a layered permeability structure. Due to large vertical offsets and regional traps for oil and gas, the north-trending Nemaha fault is a low-permeability barrier to flow across its strike in our model[23]. At the large scale of our model (145,000 km$^2$), the best-fit Arbuckle Group permeability was found to be $10^{-12}$ m$^2$, although we tested models with permeability as low as $10^{-14}$ m$^2$ (Supplementary Table 2). This large-scale bulk permeability honours the observation that >60% of wells in the region inject fluid at near-zero wellhead pressure. More importantly, after 6 years of high-volume injection throughout the region, the fluid pressure remains sub-hydrostatic, even near wells injecting tens of thousands of m$^3$ day$^{-1}$. Our modelled Arbuckle permeability predicts reservoir pore pressure changes mostly within the range between 0.1 and 2 MPa (Supplementary Fig. 2).

Because of the high permeability of the Arbuckle Group and the wide distribution of injection wells, pressure spreads out quickly and diffuses down into the much lower-permeability crystalline basement, triggering earthquakes on critically stressed faults. The earthquakes are observed at depths that typically range from 4 to 9 km[24,25]. The vertical diffusion of pore pressure is controlled by basement permeability and reflected in the characteristic time delay of several months between changes of injection and earthquake rates in the region (Fig. 1b).

We initially assumed that the permeability of the fractured basement is $10^{-16}$–$10^{-14}$ m$^2$, a range consistent with direct measurements, modelling and induced seismicity in the crystalline basement at other locations around the world[26–28]. To test this, we randomly distributed 25,000 seed points (in the study area shown in Fig. 1a) at a depth of 6.5 km below the surface. These points represent potential locations of preexisting basement faults which, if reactivated by injection-induced pressure increase, produce $M \geq 3$ earthquakes. Following the concept that changes in earthquake activity are caused by changes of the stressing (pressure) rate[29,30], we vary the basement permeability in our model and compare the rate of pressure increase (averaged over all points) to the overall observed earthquake rate ($M \geq 3$). We find that the pressure rate at seismogenic depths is most sensitive to basement permeability (see Supplementary Fig. 3). Pressure rate (and rate of pressure increase) in this publication always refers to monthly pressure accumulation at a given point in space. A basement permeability of $2 \times 10^{-15}$ m$^2$ results in the best fit between the shape of observed seismicity rates and modelled pressure rates (see Fig. 1b), consistent with the value range cited above.

Our model computes pressure changes in both time and space. We find that an open-system hydrogeologic model and high

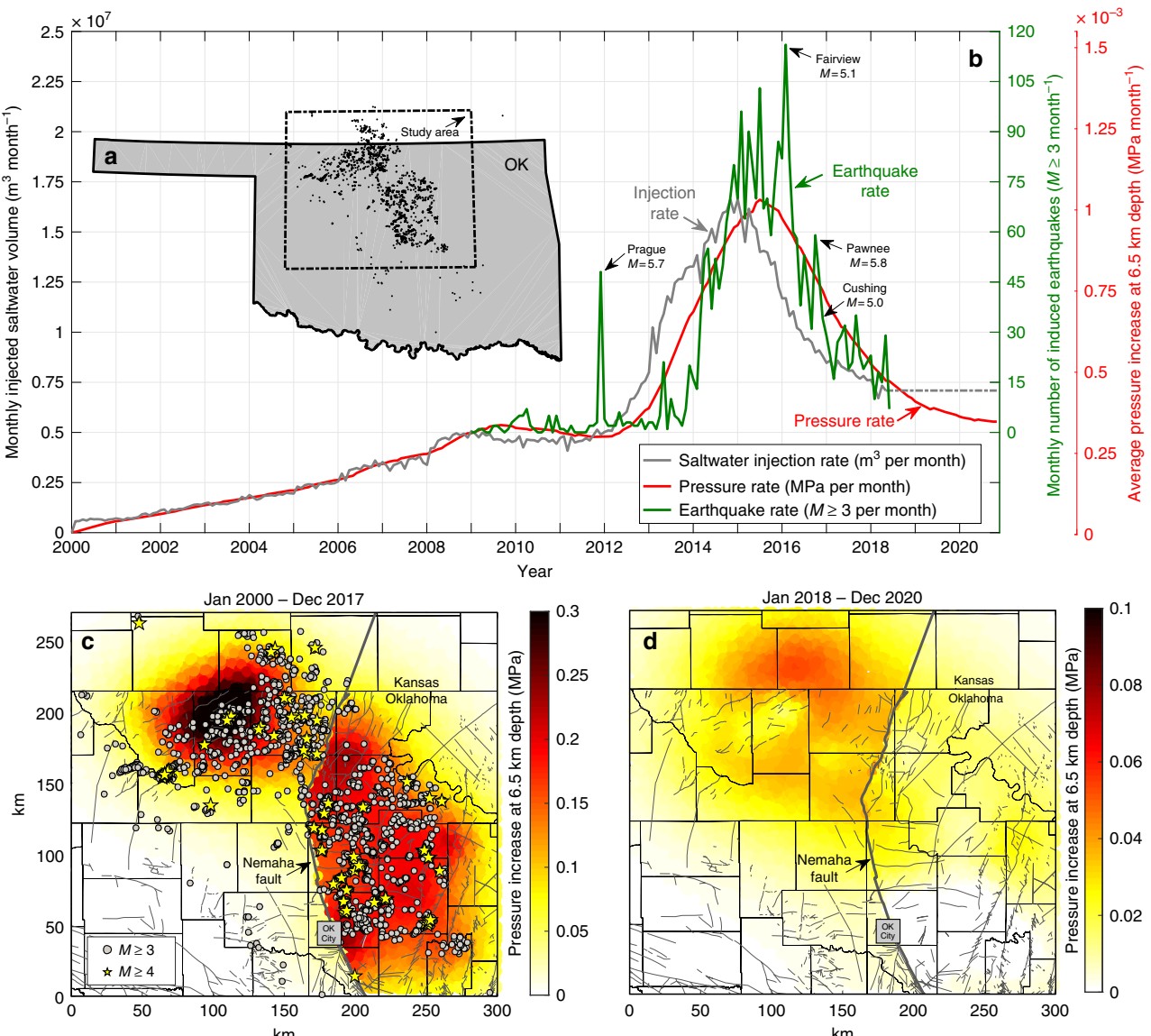

**Fig. 1** Saltwater injection, induced earthquakes and pressure increase in Oklahoma and Kansas. **a** The study area is shown by the dashed line. Black dots show $M \geq 3$ earthquakes from Jan 2009 through May 2018. **b** Total monthly saltwater injection rate (grey), observed earthquake rate (green) and average modelled rate of pressure increase (red) in the study area. Our diffusion model predicts increasing pressure at depth through 2020 (red). However, the rate of pressure increase is slowing down to the level observed in 2009, when seismicity began. **c** Map of injection-induced pressure increase at depth (Jan 2000–Dec 2017). Earthquakes ($M \geq 3$ grey dots, $M \geq 4$ yellow stars) generally occurred where injection increased pressure at depth. **d** Projected future pressure increase (Jan 2018–Dec 2020). With the exception of a region east of Oklahoma City, the pressure is expected to further increase through 2020. Note the different range of the colour schemes used in **c** and **d**. Mapped faults in the sedimentary cover[52] (**c** and **d**) are shown as grey lines

Arbuckle permeability is essential to relate seismicity rates to pressure increase. In some local-scale regions in the study area, earthquake rates and injection rates are completely unrelated and time lags of up to 10 years between peak injection and peak seismicity rates are observed (see Supplementary Fig. 4). While local-scale injection and seismicity is unrelated, the observed peak seismicity rates occur at peak pressure rates resulting from our model. Far-field pressurization, caused by high-rate injection wells outside of the considered areas, dominates and explains the high time lags between local injection and seismicity.

The map view of the cumulative injection-induced pressure increase (Jan 2000 through Dec 2017) in the complete study area shows that earthquakes generally occurred where injection increased pressure at depth (Fig. 1c). However, we find that the

seismogenic response to a given pressure increase is variable in space (Fig. 1c and Supplementary Figs. 4, 5). In some areas, a relatively small pressure increase results in a high number of earthquakes, while other regions of higher pressure increase show only a low level of seismicity. Such variations are not unexpected, because the seismogenic state, the number and stress state of preexisting basement faults affected by the pressure increase, can vary from one region to another. Based on our observations, we consider a spatially variable seismogenic state in our hybrid physics-based statistical model presented in the next section.

We use the hydrogeologic model to project future pressure changes assuming constant injection rates after March 2018 (Fig. 1b; dashed grey line). Our diffusion model predicts that, with an exception of the area east of Oklahoma City, pressure at

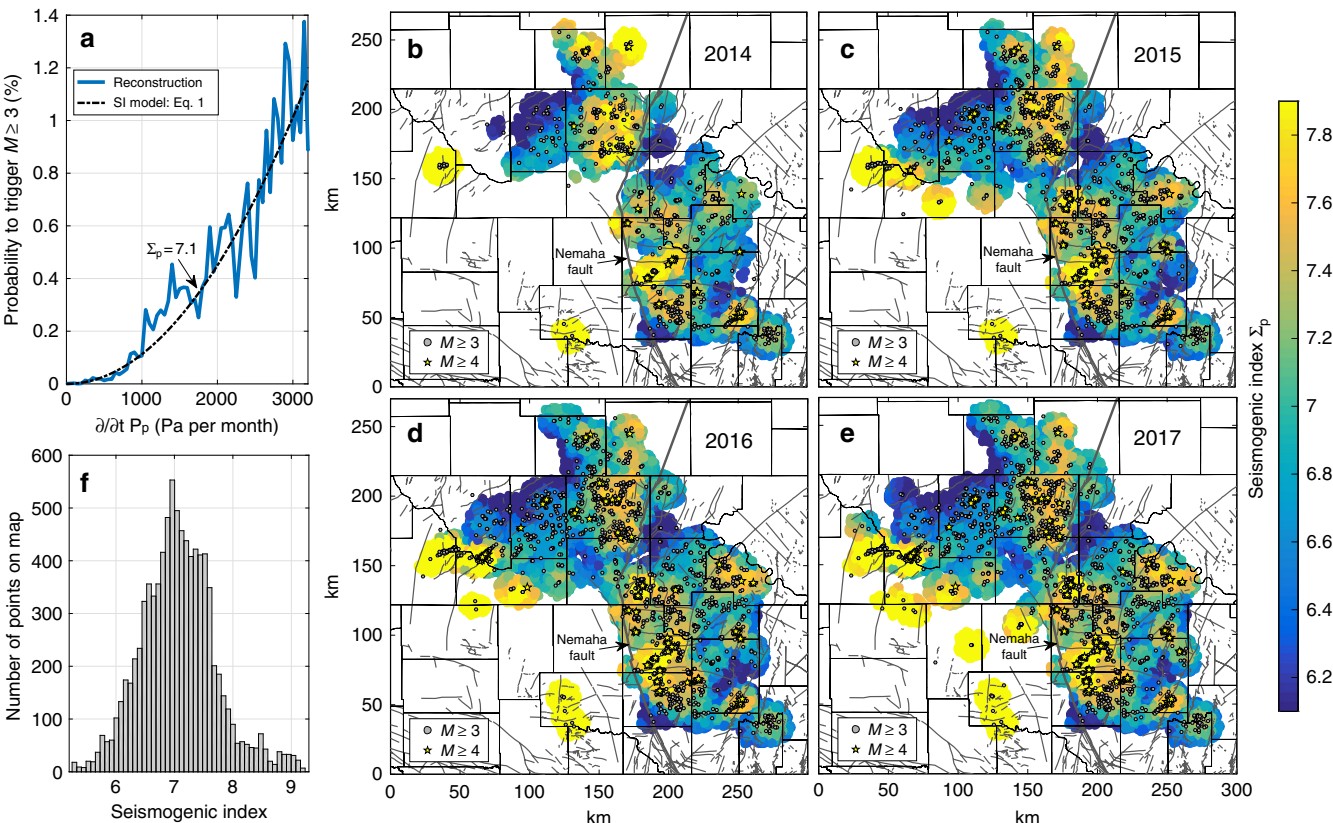

**Fig. 2** Reconstructed earthquake-triggering probability ($M \geq 3$), seismogenic index maps and histogram of spatial SI fluctuations. **a** Reconstructed probability to trigger an $M \geq 3$ earthquake at a given pressure rate in our model (see Methods for details). The reconstruction is based on observed earthquakes and modelled pressure rates through Dec 2017 in the complete study area. The probability increases with the square of the pressure rate and a SI of 7.1 (see Eq. 1). **b**–**e** Mapped spatial variability of the SI in north-central Oklahoma and southern Kansas. The SI has been computed in local-scale regions of 10-km radius around the 25,000 seed points and is calibrated through different temporal endpoints between December 2014 and December 2017. The SI shows significant variability in space but it is stable in time. Regions of high and low SI can already be differentiated using earthquakes through Dec 2014 for calibration. More details on the calibration of the SI can be found in the Methods section. **f** Histogram of the calibrated SI shown on the map in panel **e**. The mapped SI fluctuates around a mean value of about 7.1

depth continues to increase through 2020 (Fig. 1d). Increasing pressure will result in further destabilization of faults and seismicity is expected to continue. However, we expect a continuously declining earthquake rate through 2020, because the overall rate of pressure increase (Fig. 1b; red line) is continuously slowing down to the levels observed in 2009, when induced seismicity began.

**Modified Gutenberg–Richter relation for induced earthquakes.** We set up a model to transfer injection-induced pressure increases in time and space into seismicity rates by reconstructing how triggering probabilities of observed earthquakes are related to modelled pressure rates. High-pressure rates (fast pressure increases) are driving faults faster towards failure by reducing the effective normal stress. Compared to low-pressure rates, more faults are expected to reach the failure stress in a given time and we expect faster pressure increases to be more likely associated with earthquakes. The relation between pressure rate and earthquake probability (Fig. 2a) reconstructed from our model (see Methods for details) confirms the expected increase of earthquake probabilities with the rate of pressure increase. Earthquake probabilities in Oklahoma and Kansas are increasing with the square of the pressure rate (Fig. 2a). Slow pressure increase is causing a disproportionally low percentage of seismicity.

To honour our observations of the increase of the earthquake-triggering probability with the square of the pressure rate (Fig. 2a) and the spatially variable seismogenic state (Fig. 1c and Supplementary Figs. 4, 5), we combine our hydrogeologic model and a modification of the seismogenic index (SI) model[7,16]. In the combined model, we describe monthly earthquake rates $R_{\geq M}(\mathbf{r}, t)$ of magnitude $M$ and larger at location $\mathbf{r}$ and time $t$ according to a modified Gutenberg–Richter law for induced earthquakes

$$R_{\geq M}(\mathbf{r}, t) = 10^{a(\mathbf{r},t)-bM} = \left[\frac{\partial}{\partial t} P_{\mathrm{p}}(\mathbf{r}, t)\right]^2 10^{\Sigma_{\mathrm{p}}(\mathbf{r})-bM} \quad (1)$$

In Eq. 1, we introduced a space- and time-dependent earthquake productivity $a(\mathbf{r}, t) = \log_{10}\left\{\left[\frac{\partial}{\partial t} P_{\mathrm{p}}(\mathbf{r}, t)\right]^2\right\} + \Sigma_{\mathrm{p}}(\mathbf{r})$, which is determined by the monthly injection-induced pressure rate $\frac{\partial}{\partial t} P_{\mathrm{p}}(\mathbf{r}, t)$ in space and time and the spatially variable SI $\Sigma_{\mathrm{p}}(\mathbf{r})$. The quantity $10^{\Sigma_{\mathrm{p}}(\mathbf{r})}$ in Eq. (1) is a proxy for number and stress state of basement faults at location $\mathbf{r}$. The higher the SI at a location, the higher the earthquake rate caused by a given pressure increase, because a higher number of (or more critically stressed) preexisting faults are affected by the pressure increase. In agreement with the classical Gutenberg–Richter[31] law, $10^{-bM}$ describes magnitude scaling in our model. The SI and the $b$-value can be calibrated based on observed earthquakes and modelled

pressure rates up to a given time. The calibrated parameters and modelled future pressure rates can then be used to forecast expected earthquake rates above a given magnitude in space and time according to Eq. (1).

Note that the classical SI model[16] considers injection rates instead of pressure rates and is not (directly) applicable to decreasing injection rates. While Langenbruch and Zoback[7] bypassed this problem by considering a modification of Omori's law[32] to describe the decay rate of seismicity after injection rates start to decrease, in this study, we replace injection rates by pressure rates utilizing a hydrologic model to make our model applicable to arbitrary injection scenarios.

**Mapping the seismogenic state.** To characterize spatial variability of the seismogenic state throughout the study area, we calibrate $\Sigma_p(\mathbf{r})$ in local-scale regions of 10-km radius around the 25,000 seed points (Fig. 2b–e) by analyzing pressure changes and earthquake rates in each area (see Methods for details). Note that the SI in our model is determined based on observed earthquakes and modelled pressure increases at the fault seed points and is not directly comparable to values determined in other studies using injection volumes[7,16].

We find that $\Sigma_p(\mathbf{r})$ varies by about two units across the study area (Fig. 2b–f). This would mean that in areas of the highest SI, a given pressure increase will cause one hundred times as many earthquakes as in areas characterized by the lowest SI. The SI maps (Fig. 2b–f) can be interpreted as a map of critically stressed faults in the crystalline basement in the areas affected by pressure changes. The areas of highest SI imply one hundred times as many preexisting faults than the areas of lowest SI. Thus, both the local pressure increase and the local seismogenic state must be combined to determine the local rate of induced earthquakes.

To test the concept that spatial variations of $\Sigma_p(\mathbf{r})$ reflect the susceptibility of preexisting faults (formed over geologic time) to pressure changes, we show in Fig. 2b–e that SI maps calibrated through different time periods show only minor fluctuations. This suggests that as soon as the local SI can be determined from local seismicity, Eq. (1) can be used to forecast the expected rate of earthquakes caused by future pressure increase in the considered region.

**Physics-based seismic hazard maps.** Using our model, we produce 1-year maps of the seismic hazard to assess the probability of potentially damaging induced earthquakes in Oklahoma and Kansas from 2015 through 2020. To produce real forecasts based on knowledge about future injection rates, we only use observed earthquakes and modelled pressures through the end of a given year to calibrate the model (SI map and b-value) (Fig. 2b–e and Supplementary Fig. 7). According to Eq. 1, we then use modelled pressure rates in the coming year to forecast expected seismicity throughout the study area.

Note that fluctuations of the b-value calibrated through different times between Dec 2014 and Dec 2017 are within the uncertainty of the computation (see Supplementary Fig. 7). A spatially constant b-value is applied in the complete area, because b-values computed for small regions, which include a lower number of earthquakes, would show even larger uncertainties and have a high likelihood of being artefacts[33].

Considering that the occurrence of induced earthquakes is a Poisson process[7,13,16,34] the expected annual number ($N_{\geq M}$) of earthquakes of magnitude $M$ or larger can be computed corresponding to Eq. 1 and the probability to exceed magnitude

$M$ per year can be determined according to

$$Pr(M) = 1 - Pr(0, M, N_{\geq M}) = 1 - \exp(-N_{\geq M}) \qquad (2)$$

$Pr(0, M, N_{\geq M}) = \exp(-N_{\geq M})$ corresponds to the probability that no earthquake of magnitude $M$ or larger occurs in 1 year, if $N_{\geq M}$ earthquakes of magnitude $M$ or larger are expected.

We choose a magnitude of $M = 4$, because lower magnitudes are very unlikely to cause damage. The resulting prospective 1-year maps of the probability to exceed $M = 4$ (Fig. 3) show how the seismic hazard in time and space is changing in response to spatial and temporal variations of injection rates and spatial changes of the SI. The probabilities shown in Fig. 3 were computed in regions of 20-km radius around the 25,000 seed points. However, the hazard maps can be determined for arbitrary magnitude thresholds, regions and time scales.

We find that earthquakes observed in the year of the predictions occur where our model forecasts enhanced exceedance probabilities (Fig. 3a–d). In total, 64 of 65 $M \geq 4$ earthquakes recorded from Jan 2015 through May 2018 occurred where our model predicts annual exceedance probabilities above 10%. In total, 57 of 65 $M \geq 4$ earthquakes occurred within contours of 30% exceedance probability.

In response to decreased saltwater injection rates, pressure increases are slowing down over a wide range of depths (Supplementary Fig. 8) and our model forecasts a widespread reduction of the seismic hazard in 2017. Even without additional injection rate reductions after March 2018, our model predicts a further decrease of the seismic hazard in 2018, 2019 and 2020. Note that this decrease is caused by the diffusive nature of the pressure migration and the disproportionally low probability of earthquake triggering for slow pressure increases.

East of the Nemaha fault, where injection rates have been reduced most significantly, the strongest decrease is predicted (see Fig. 3). In other parts of northern Oklahoma and southernmost Kansas, $M \geq 4$ probabilities remain on a higher level. Our physics-based maps identify three regions where $M \geq 4$ exceedance probabilities in 2018 remain above 30% (Fig. 3d). Note that most earthquakes observed in 2018 to date occur in or close to these regions. Earthquake probabilities in 2018, 2019 and 2020 are based on the assumption of a constant injection level after March 2018. Further injection rate reductions would accelerate the probability decrease and our model can be updated as soon as new injection data become available.

More importantly, a significant advance of our physics-based method is that it can be used to identify the optimal injection strategy to mitigate the remaining seismic hazard. Alternative scenarios for future injection rates in space and time could easily be considered to evaluate how possible injection regulations would affect the seismic hazard. Not only further reduction in total injection volume, but also a redistribution of injection volumes in the study area could mitigate the seismic hazard. Moving injection volume away from critical regions of high SI could lower the injection-induced seismic hazard without reducing the overall volume of injection.

**Retrospective performance evaluation in time and space.** To retrospectively evaluate the performance of our model, we compare the observed seismicity rate ($M \geq 3$) in the complete study area to the forecasted seismicity rate resulting from SI models calibrated through different temporal endpoints. We select a magnitude threshold of $M = 3$, because the model performance should be tested based on the highest possible number of observations. Note that while the observed rate of $M \geq 4$

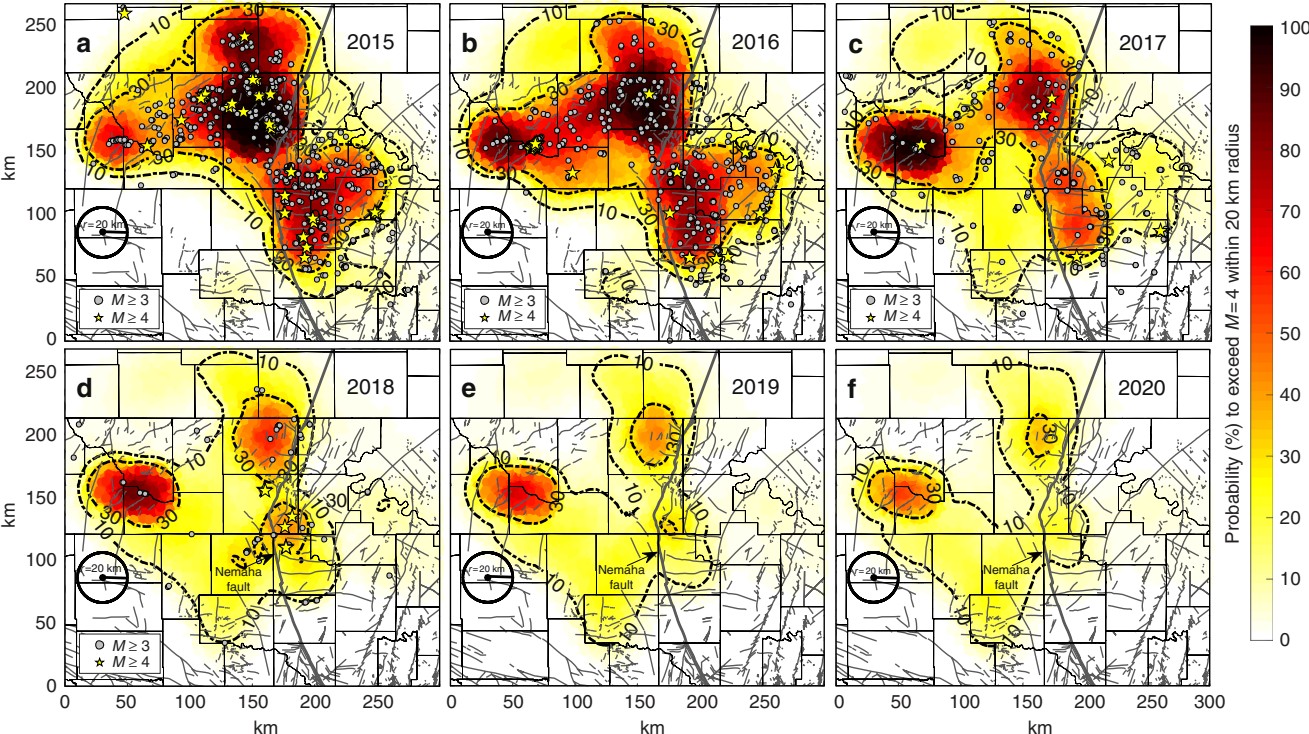

**Fig. 3** Physics-based 1-year magnitude ($M \geq 4$) exceedance probability forecasts (2015–2020). Exceedance probabilities are forecasted in areas of 1257 km$^2$ (20-km radius) and for the time of 1 year. Grey circles and yellow stars show $M \geq 3$ and $M \geq 4$ earthquakes observed in the year of the forecast, respectively. The local seismic hazard is controlled by local pressure increase at depth and the local seismogenic state (the SI Fig. 2b–e). Decrease of magnitude probabilities from 2015 to 2020 is driven by reduced injection rates, which slow down the pressure increase at seismogenic depth. The strongest decrease of the seismic hazard is predicted east of the Nemaha fault, where injection rates were reduced most significantly. Probabilities of 2018–2020 were computed assuming constant injection rates after March 2018. The maps can be produced for arbitrary future injection scenarios to optimize the distribution of fluid injection volumes in space and time for seismic hazard mitigation

earthquakes falls well within the uncertainty range of our model (Supplementary Fig. 9), it does not allow us to draw statistically significant conclusions, because of the high uncertainty caused by the smaller number of $M \geq 4$ observations.

The modelled seismicity rate forecasts of $M \geq 3$ (Fig. 4a) reproduce increase, peak and decrease of the overall observed earthquake rate. However, some short-term spikes peak out of the expected range of seismicity. These spikes are caused by aftershocks of the relatively larger earthquakes such as the $M = 5.7$ Prague, $M = 5.1$ Fairview and $M = 5.8$ Pawnee earthquakes. In the same way, the slightly under-predicted peak of seismicity in 2015 consists of various spikes of aftershocks following earthquakes of moderate magnitudes (Supplementary Figs. 4, 5). Our model relates long-term trends of produced-water injection to long-term trends of seismicity and is not designed to forecast short-term spikes related to aftershocks. We decided not to remove aftershocks from the catalogue, because declustering depends on a subjective algorithm[35,36] and parameter choices. Moreover, aftershocks contribute to the seismic hazard. Removing them from a hazard assessment seems inappropriate[15].

Figure 4a demonstrates that changing the temporal endpoint of the calibration between Dec 2011 and Dec 2017 has no significant effect on forecasted seismicity rates. Had we only used earthquake information through Dec 2011 to calibrate the SI, our model would have successfully predicted the increase, peak and decrease of the large-scale seismicity rate in response to changes of injection rates.

Ultimately, the most important application of our model is that it can be used to predict local-scale seismicity rates, because the SI is calibrated on the local scale. In Supplementary Fig. 5, we

present seismicity rate forecasts in six local-scale study areas of 25-km radius. The areas are selected to represent the full value range of the SI identified across the complete study area (Supplementary Fig. 6). Our results show that our model can be used to forecast local-scale seismicity rates. However, because the local-scale SI can vary significantly from the large-scale average, local-scale model calibration (based on local seismicity and pressure increase) is required to successfully forecast local-scale seismicity rates. We find that compared to the purely observational approach of using last year's earthquakes to predict next year's seismicity, incorporating injection rates and a spatially variable SI through our physics-based approach significantly increases the forecasting performance in time and space (Supplementary Fig. 10).

In Fig. 4a and Supplementary Fig. 5, we compare our results to a recent paper[37] which also considers injection rates to model seismicity rates in Oklahoma and Kansas based on rate and state friction[38]. It considers the Arbuckle group as a closed hydrogeologic system with equally distributed injection rates throughout the region. Because no forecast of the seismicity rate (2018–2020) is presented in the publication, we forecast the seismicity rate through Dec 2020 based on the rate-and-state model[37] assuming constant injection rates after March 2018 as in our analysis.

While the large-scale seismicity rates resulting from the rate-state and the SI models are similar from 2012 through 2015, the rate-state model significantly under-predicts the decay rate following reduced injection rates (Fig. 4a). Moreover, as the rate-and-state model does not consider spatial variability of model parameters, it is not able to describe onset, increase, peak

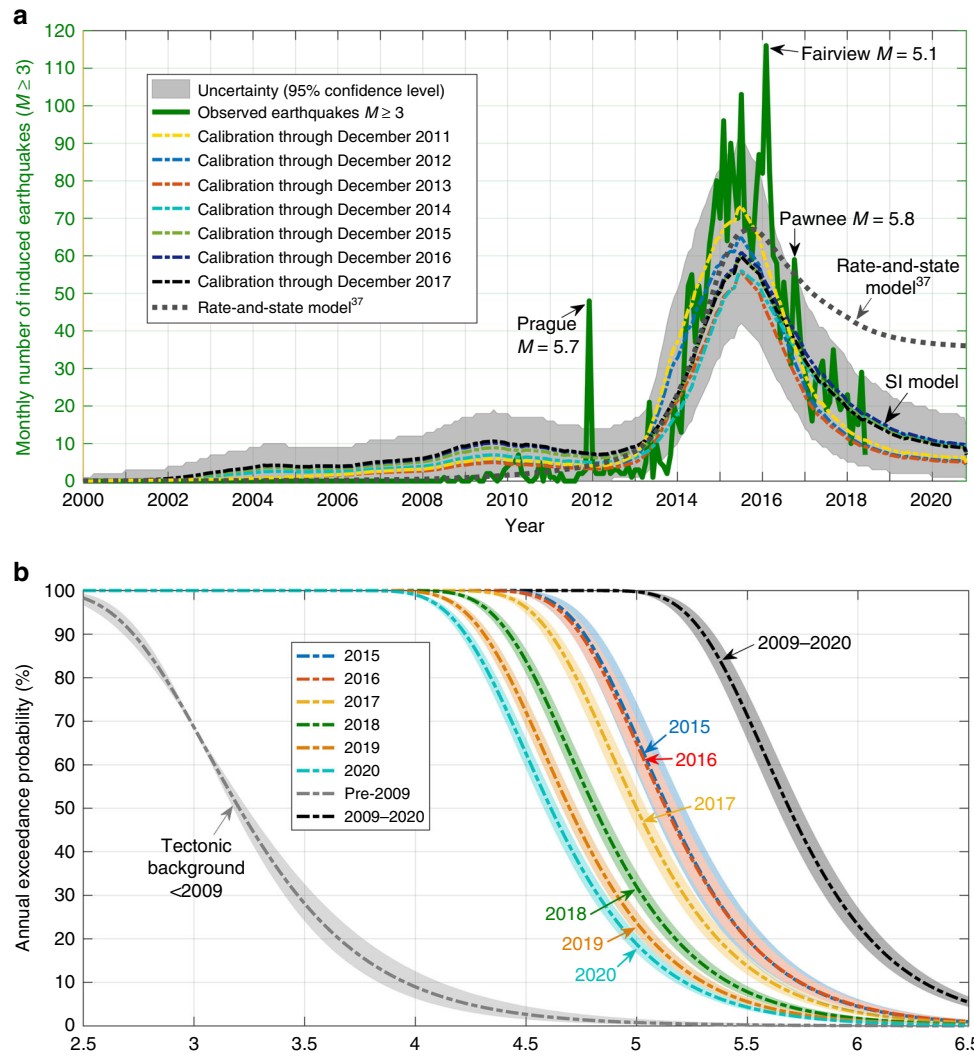

**Fig. 4** Forecasted seismicity rate ($M \geq 3$) and magnitude exceedance probability in the complete study area. **a** Observed earthquake rate (solid green line), and forecasted seismicity rates resulting from SI models calibrated through different temporal endpoints (Dec 2011–Dec 2017) (coloured dashed lines). Changing the temporal endpoint of the calibration has only a minor effect on forecasted seismicity rates. A recently published model[37], based on rate and state friction, significantly under-predicts the decrease of the earthquake rate caused by reduced injection (dotted grey line). **b** Forecasted annual exceedance probability of magnitude M in the complete study area. The forecasted probability of potentially damaging earthquakes is significantly enhanced compared to the tectonic background and is the highest in 2015 and 2016. Background probabilities are adopted from Langenbruch and Zoback, 2016[7]. In response to reduced injection rates magnitude exceedance probabilities are decreasing, but still above the tectonic background level through 2020. The overall injection-induced pressure increase from 2009 to 2020 has a potential of 23% to trigger a damaging $M \geq 6$ earthquake (6% of $M \geq 6.5$). Uncertainties (shaded areas) represent $b$-value computations (see Supplementary Fig. 7)

or decay of seismicity in any of the six local-scale regions presented in Supplementary Fig. 5.

## Discussion

Based on a physical understanding of injection-induced earthquakes, we developed forecasts of the spatially variable and time-dependent earthquake hazard in Oklahoma and Kansas. In our hybrid physical–statistical model, occurrence probabilities of potentially damaging earthquakes are driven by two factors: (1) the rate of injection-induced pressure increase at depth resulting from the locations and injection rates of the saltwater disposal wells and (2) spatial variations in number and stress state of preexisting basement faults that are affected by the pressure increase.

Other studies[16,32] observe a direct proportionality between pressure increase (injection rates) and earthquake probability during hydraulic stimulation of enhanced geothermal systems and hydraulic fracturing sites. However, we find that earthquake probabilities in Oklahoma and Kansas are increasing with the square of the pressure rate (Fig. 2a). Slow pressure increases cause a disproportionally low percentage of seismicity. Relatively more aseismic fault creep could be occurring when there are slow pressure increases. Moreover, aftershock triggering might contribute to the over-proportional increase of probabilities with the rate of pressure increase.

Our finding of disproportionally low earthquake probabilities at slow pressure increases has important implications for hazard management of induced earthquakes. It underlines the effectiveness of injection rate reductions, which slow down the rate of

pressure increase at seismogenic depth. At the current level of saltwater injection, our model forecasts slowly decreasing earthquake probabilities through 2020. As shown in Fig. 4a, about 190, 130 and 100 widely felt $M \geq 3$ earthquakes are expected in 2018, 2019 and 2020, respectively. This represents a significant reduction compared to 943 earthquakes of $M \geq 3$ observed in 2015.

When communicating probabilistic seismic hazard forecasts to decision-makers or the public, it is important to emphasize that forecasted probabilities strongly depend on the considered scale in time and space. For instance, the annual probabilities to exceed $M = 5$ in 2018, 2019 and 2020 are low ($\leq 5\%$) in local-scale regions of 20-km radius (Supplementary Fig. 11). However, the annual probability that there will be an $M \geq 5$ somewhere in the affected area is as high as 32%, 24% and 19% in 2018, 2019 and 2020, respectively (Fig. 4b).

Also a longer timescale of interest increases the forecasted probability of earthquake damage. Considering the cumulative injection-induced pressure increase from 2009 to 2020 results in a 24% chance of a damaging $M \geq 6$ (6% of $M \geq 6.5$) earthquake (Fig. 4b). Based on the predicted pressure increase from Jan 2018 through Dec 2020, one additional $M \geq 5$ earthquake is expected to occur (Supplementary Fig. 12). Seismic activity is expected to remain elevated in respect to the tectonic background of about one $M \geq 3$ earthquake and a probability of about 1% to exceed $M \geq 5$ per year[7] observed in the region prior to 2009. Thus, as long as earthquakes are induced by saltwater injection in Oklahoma and Kansas, occurrence of even larger magnitudes (than those observed to date) cannot be ruled out completely. To further mitigate the remaining induced seismic hazard alternative, future injection scenarios could be used as input to our method to identify the optimal injection strategy in time and space.

## Methods

**Injection well data**. We specifically analyzed and utilized Oklahoma Corporation Commission (OCC)[39] and Kansas Corporation Commission (KCC) injection data for Arbuckle Group injection wells operating in the area of interest[12] in north-central Oklahoma as well as Harper and Sumner counties in south-central Kansas. We obtained injection data for wells in Oklahoma and Kansas going back as far as January 2000. Only a handful of wells contain injection information back that far, but we used the most complete dataset of injection data to simulate the largest possible build-up of pressure in the model.

Monthly injection data for most Kansas wells began to be reported in January 2012. Prior to the reporting of monthly injection data, we assumed that annual injection was distributed proportionally across all 12 months of the year. Injection data in Oklahoma and Kansas were collected through March 2018 and December 2016, respectively. All models which project future trends in pressure and seismicity assume that injection wells operated at constant rates are equal to the average of the last three reported months of injection data.

**Numerical hydrogeologic model**. Our numerical model of the hydrogeologic system of Oklahoma's deepest sediments and crystalline basement is aimed at predicting fluid injection-related pore pressure changes at depth in space and time. Here, we developed a three-dimensional hydrogeologic model which simulates fluid injection from wells operating in the Arbuckle Group from January 2000 through December 2020. We simulate the diffusion of pore pressure using MODFLOW, a modular finite difference numerical code developed by the U.S. Geological Survey[40]. MODFLOW solves for changes in hydraulic head, which are directly proportional to pore pressure, using the groundwater flow equation

$$\frac{\partial}{\partial x}\left(K_{xx}\frac{\partial h}{\partial x}\right) + \frac{\partial}{\partial y}\left(K_{yy}\frac{\partial h}{\partial y}\right) + \frac{\partial}{\partial z}\left(K_{zz}\frac{\partial h}{\partial z}\right)$$
$$= S_s\frac{\partial h}{\partial t} - Q_i(t)\delta(x-x_i)\delta(y-y_i)\delta(z-z_i), \qquad (3)$$

where $h$ is the hydraulic head [L], $K_{ij}$ are the principal components of the hydraulic conductivity tensor [LT$^{-1}$], $S_s$ is the specific storage coefficient [L$^{-1}$] and $x$, $y$, $z$ and $t$ are spatial and temporal coordinates. $Q_i$ is the fluid source or sink (T$^{-1}$). Changes in hydraulic head are directly proportional to changes in pore pressure when accounting for the specific weight of water ($\Delta P_p = \rho g \Delta h$). The hydraulic conductivity tensor is directly proportional to permeability ($k$) when accounting both for the specific weight ($\rho g$) and dynamic viscosity ($\mu$) of the fluid ($K_{ij} = k_{ij}\frac{\rho g}{\mu}$). The numerical code assumes that injected fluids are of constant density and dynamic viscosity. The fluid density and fluid dynamic viscosity in the model are 1062 kg m$^{-3}$ and 0.547 cP, respectively (Supplementary Table 1). To calculate the

fluid density and dynamic viscosity used in the model, we assume a brine TDS of 100,000 ppm[41] and a reservoir temperature of 50 °C derived from a standard geothermal gradient of 25 °C km$^{-1}$.

Injection in our model occurs into the 2.1-km-deep, 400-m-thick Cambrian-aged, fractured dolomitic carbonate Arbuckle Group and is assumed to be uniform across the entire depth of the interface. The Arbuckle Group is underlain across the entire model domain by a lower-permeability crystalline basement to a depth of 20 km (Supplementary Fig. 13). We explicitly calculate all pressure changes in the model relative to a uniform, hydrostatic pre-injection baseline. The model domain stretches 406 km from east to west and 357 km from north to south, as measured from the model's northwest corner at UTM 409865 E, 4186546 N Zone 14. All model boundaries are of Neumann type (no-flow). Model boundaries in $x$ and $y$ were set intentionally far from the dominant region of injection such that they have no effect on pressure calculations. The upper boundary is reflective to the confined condition of the Arbuckle Group reservoir[8]. The lower boundary of the model domain is set intentionally very deep so as to have no effect on pressure calculations at hypocentral depths. Grid discretization in $x$–$y$ is 500 × 500 m$^2$ with some variation as the outer edges of the model are approached. The model contains 12 discretization layers in the $z$-direction to accurately capture the diffusion of pore pressure in the vertical direction (Supplementary Fig. 13).

Our hydrogeologic model intentionally simplifies and idealizes the three-dimensional hydrogeologic system's large-scale bulk permeability with uniform layered heterogeneity. The initial model parameterizations are based on the reported range of Arbuckle Group permeability, $10^{-14}$–$10^{-12}$ m$^2$, from core, outcrop and monitoring well tests[18–22]. While lacking in direct measurements of crystalline basement permeability in Oklahoma and southern Kansas, we implemented bulk permeabilities of fractured basement rock from the literature ($10^{-16}$–$10^{-14}$ m$^2$)[27,28,42]. Due to the inherent uncertainty in model parameterizations, we tested the sensitivity of model outputs to permeability by running several combinations of Arbuckle Group and crystalline basement permeabilities (Supplementary Table 2).

Specific storage in our model is a parameter derived from the porosity, bulk compressibility of water and bulk compressibility of rock reported in the literature for the Arbuckle Group and crystalline basement (Supplementary Table 1). Specific storage combines the bulk compressibility of water ($\beta_w$), bulk compressibility of rock ($\beta_r$), fluid density ($\rho$) and porosity ($\Phi$) as described below:

$$S_s = \rho g(\beta_w \Phi + \beta_r), \qquad (4)$$

We use a fluid density of 1062 kg m$^{-3}$ reflective of brine with TDS of 100,000 ppm[41] at a reservoir temperature of 50 °C. The bulk compressibility of water of $4.4 \times 10^{-10}$ Pa$^{-1}$ respects the reservoir temperature. The bulk compressibility and porosity of the Arbuckle Group is $0.16 \times 10^{-10}$ Pa$^{-1}$ and 0.20 derived from direct observations and core testing[20,22,43]. The bulk compressibility and porosity of the crystalline basement is set from literature value to be $0.7 \times 10^{-11}$ Pa$^{-1}$ and 0.01, respectively[44]. All of our model sensitivities use specific storage of $1 \times 10^{-6}$ m$^{-1}$ and $1 \times 10^{-7}$ m$^{-1}$ for the Arbuckle Group[22,45] and crystalline basement[44], respectively.

Beyond the layered heterogeneity represented in our model, a key feature in our model is the representation of the Nemaha fault as a regional-scale, low-permeability barrier to cross-fault flow ($k = 10^{-20}$ m$^2$). The Nemaha fault is a structural uplift of Pennsylvanian–Permian age which runs roughly north–south from central Oklahoma through Kansas[23]. The structural uplift has served as a regional trap for oil and gas structures such as the Oklahoma City oil field[46,47].

Two key baseline datasets constrain the large-scale Arbuckle Group permeability: (1) the observed hydraulic underpressure for the deepest hydrostratigraphy in Oklahoma and Kansas, and (2) large-scale trends in reported daily wellhead pressures for Arbuckle injection wells in Oklahoma. The observed hydraulic underpressure, or the difference between the ambient fluid pressure and the land surface elevation, in aquifers of Cambrian–Ordovician–Silurian age in the U.S. mid-continent is well-known[41]. We used hydro-potentiometric surface data for aquifers of Cambrian–Ordovician–Silurian age from Nelson et al. (2015)[41], which were derived from drill-stem tests and calculated hydraulic underpressures at Arbuckle injection well locations (Supplementary Fig. 2a).

In early 2016, the Oklahoma Corporation Commission began mandating daily wellhead pressure measurements for Arbuckle injection wells operating within the area of interest in Oklahoma. Large-scale trends in daily wellhead pressures show that more than half of all Arbuckle injection wells, even wells operating at rates >10,000 m$^3$ day$^{-1}$, operate under gravity-feed injection requiring no wellhead pressure. Most of the remaining wells (~40%) operate at wellhead pressures between ~0.3 and 2 MPa[41], well within the pressure range expected simply from wellbore friction. These data indicate that the large-scale, bulk permeability of the Arbuckle Group is likely towards the high end of the reported range.

**Injection-induced fault reactivation probability**. To compute the probability of triggering $M \geq 3$ earthquakes in our model (Fig. 2a), we divide the histogram of pressure rates triggering observed earthquakes (Supplementary Fig. 14a) by the histogram of monthly pressure rates at all 25,000 seed points in the model (Supplementary Fig. 14b). Pressure rates triggering observed earthquakes are extracted in the month of occurrence and at the epicentre locations of all $M \geq 3$ earthquakes in the catalogue through Dec 2017 (depth is fixed at 6.5 km below the surface).

The histogram of earthquakes triggered at a given pressure rate (Supplementary Fig. 14a) does not directly give the probability to trigger an $M \geq 3$ earthquake (fault seed point) in our model. For instance, 90% of all pressure rates in the model are <1000 Pa month$^{-1}$, but only 15% of earthquakes are triggered in this range (Supplementary Fig. 14b).

Our model suggests that injection-induced pressure rates at seismogenic depths are low. Ninety-nine percent of all pressure rates at the 25,000 fault seed points are below 3000 Pa month$^{-1}$ (Supplementary Fig. 14b). While our modelled rates of pressure increase are low, they fall within the range that has been observed to trigger natural earthquakes. Inter-seismic Coulomb stressing rates on the San Andreas Fault[48,49], for instance, are found to be on the order of 1000 Pa month$^{-1}$.

**Calibration of the seismogenic index.** Before calibrating the SI maps (Fig. 2b–e), we precondition the pressure rates resulting from the hydrogeologic model based on the physical understanding of pore pressure changes as the triggering mechanism of induced earthquakes. (1) Only positive pressure rates (pressure increases) are considered in our model, because decreasing pressure results in fault strengthening by increasing the effective normal stress. (2) If the pressure at a given seed point at a considered time is lower than the maximum observed pressure at any earlier time, the pressure rate is not considered. We do this to honour the stress memory effect of rocks (Kaiser effect), describing the necessity to exceed the previous observed maximum pressure (stress) level before seismicity is observed[50,51]. Our results show that including this effect has no large influence on calibrating the SI in Oklahoma and Kansas but might be important for application to other areas, where injection rates show a cyclic behaviour.

The following steps are performed to calibrate the SI maps (Fig. 2b–e) by comparing earthquake and pressure history in circular areas of 10-km radius. (1) Monthly pressure rates $\frac{\partial}{\partial t} P_{\mathrm{p}}(\mathbf{r}_n, t)$ at all $n$ seed points within a radius of 10 km around a selected seed point up to a given calibration time $t_c$ are extracted, squared and summed up to $\sum_n \left[\frac{\partial}{\partial t} P_{\mathrm{p}}(\mathbf{r}_n, t \leq t_c)\right]^2$. (2) The number $N_{M \geq 3}(t \leq t_c)$ of $M \geq 3$ earthquakes within 10-km radius around the selected seed point observed up to the given calibration time is determined. (3) The maximum likelihood estimate of the $b$-value is computed using all $M \geq 3$ earthquakes recorded through the calibration time in the complete study area (Supplementary Fig. 7). (4) The SI at location $\mathbf{r}$ is determined according to

$$\Sigma_{\mathrm{p}}(\mathbf{r}) = \log_{10}\left\{N_{M \geq 3}(t \leq t_c)\right\}$$
$$- \log_{10}\left\{\sum_n \left[\frac{\partial}{\partial t} P_{\mathrm{p}}(\mathbf{r}_n, t \leq t_c)\right]^2\right\} + b(t_c)M. \tag{5}$$

We find that as soon as two earthquakes occurred in the local-scale areas of 10-km radius around a selected seed point, a good estimate of the SI can be obtained. Note that $b$-values are determined based on earthquakes in the complete study area. Where the number of earthquakes is not sufficient (<2 earthquakes) to directly calibrate the SI based on observed earthquakes and modelled pressure rates, we apply the following rule to obtain a SI value. If no directly calibrated SI value exists within 40-km radius around a selected seed point, the SI is set to the mean value of all directly calibrated SI values. If directly calibrated SI values exist within 40-km radius, the SI is interpolated (using the *griddata* function in Matlab). Dotted areas in Supplementary Fig. 6 show areas where interpolated or averaged SI values are used to create the seismic hazard maps shown in Fig. 3.

## Data and code availability

Earthquake catalogue: http://earthquake.usgs.gov/earthquakes/search/. Injection well location and saltwater injection data: http://www.occeweb.com/og/ogdatafiles2.html. Additional data and codes supporting the findings of this manuscript are available from the corresponding author upon reasonable request.

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

## Acknowledgements

This study would not have been possible without the Underground Injection Control injection data provided by the Oklahoma Corporation Commission (OCC) and the Kansas Corporation Commission (KCC) and the National Earthquake Information Center earthquake catalogue provided by the U.S. Geological Survey. We thank Justin Rubinstein for providing the digitized injection data obtained from the KCC. The Stanford Center for Induced and Triggered Seismicity provided the financial support for this study.

## Author contributions

C.L. conceived the study and performed the analysis except the hydrogeologic modelling part. M.W. implemented and performed the hydrogeologic modelling. M.D.Z. contributed to continual supervision and guidance; C.L. created all figures (except Supplementary Figs. 2, 13 created by M.W.), wrote and revised the manuscript with substantial input from M.W. and M.D.Z.

## Additional information

**Competing interests:** The authors declare that they have no competing interests.

