## [Peer Review File · Nature Communications]

Reviewers' comments:

Reviewer #1 (Remarks to the Author):

Peer Review Comments for
Physics-based forecasting of man-made earthquake hazards in Kansas
By C. Langenbruch, M. Weingarten, and M. Zoback

Scope of Review: My expertise is in the field of nonisothermal, multi-phase fluid flow in porous and fracture geologic media with a specific emphasis on modeling subsurface fluid injections. Consequently, the scope of this review is limited to the *fluid* aspects of fluid-triggered earthquakes. I am not qualified to provide rigorous peer-review in the area of seismological processes.

General Comments: This manuscript presents a novel, physics-based method to estimate spatiotemporal variations of fluid-triggered earthquakes using north-central Oklahoma and southern Kansas as the study area for development and testing. In this study, the authors modify the Seismogenic Index calculation to account for deep underground injections of oilfield wastewater, and the resulting fluid pressure changes that are known to trigger earthquakes. The novel approach taken in this study is designed to ameliorate the problem of non-stationarity in calculations of fluid-triggered earthquake hazard, which traditionally assumes a constant earthquake rate. Overall this is a timely and provocative idea, and one that is ideally suited for rapid publication in a widely read scientific journal, such as *Nature-Communications*. While I do support publishing this study, I think there is also room for improvement, particularly in the context of how the numerical groundwater model is presented to the readership. Many of the specific comments below are based on the idea that the methods must be clearly articulated in order for this method to be usable by the larger community working on fluid-triggered earthquakes. For example, a number of comments relate to more clearly presenting the model geometry, initial/boundary conditions, and presenting a very clear discussion of the model assumptions, as well as the potential implications of these various assumptions on the model results. Given the massive size and scope of the groundwater modeling presented in this study, these comments should not be perceived as indication that the modeling is inadequate—in contrast, the regional-scale groundwater model is an impressive accomplishment—however, I do believe that it is critically important for the reader to understand the model assumptions & limitations. The other specific comment that is worth mentioning here is that I had some difficulty understanding the concept of “pressure rate” as it is presented in the manuscript & supplemental. I think additional discussion would be very useful to the reader in making sense of this important concept.

In conclusion, this is an important study and I support publication after modest revisions to the groundwater modeling discussion & methods.

Signed,
Ryan M. Pollyea

Specific Comments

Line 74: spatially variability → spatial variability

Lines 107 – 120: Calibrating the permeability against earthquake is an interesting approach. Was permeability anisotropy considered? Or porosity (through S_s)?

Line 114 – 115 & Fi. 1b: This study relies heavily on the concept of “pressure rate,” which is defined in Extended Fig. 5a as, dP_f/dt . After reading the primary manuscript, supplemental material, and references 28-29, I can't seem to figure out what “pressure rate” actually means in the context of the paper. Figure 1b shows that the pressure rate at 6.5 km depth reaches ~25 MPa/month in 2016 – this is a *lot* of fluid pressure, and perhaps a bit misleading because Fig 1c shows ~200 kPa of pressure accumulation across the model domain after 17 years of injection. How does 25 MPa per month only result in 200 kPa of pressure after 17 years? Is the pressure data in Fig 1b the total pressure accumulation over the complete set of seed points per time? If so, I'm not sure what this value actually means because the seed points are used as a proxy for potential fault locations, so the cumulative pressure across all seed points is less important than the local pressure accumulation at each one. Would average ΔP as a fn of time serve as a

better proxy for changes to effective stress at 6.5 km depth? This would more clearly connect changes in the overall pressure regime to the faults (seed points) that are potentially affected, while also allowing a measure of uncertainty about the pressure curve to be shown. Regardless of how the authors choose to represent the overall pressure change at seismogenic depths, I think that a more information is needed in the supplemental to clearly articulate the concept of *pressure rate*.

Figure 1b: In this figure, it is not clear what data are observed and what data are modeled. It seems reasonable that volume and earthquake data are observed/measured through 2017, and pressure data is modeled; however, this should probably be stated in the caption – particularly, because everything ≥ 2018 is simulated.

Figures 1b & 1c: These panels use the same color scheme, but different range – maybe mention the different range in the caption.

Line 212: Typo: opne \rightarrow one

Lines 293 – 295: It is clear that the hazard model developed here is driven by spatial and temporal variations in pressure accumulation at seismogenic depths, but it is less clear that the model takes into account the spatial variability of optimally aligned faults b/c the overall method incorporates pressure accumulation rate into the SI calculation, which is dependent on earthquake rate. I don't see where *a priori* knowledge about fault location/geometry is part of the model, except perhaps in the context of earthquake occurrence, but this can only be forward modeled in a probabilistic sense, where as pressure diffusion in this study is forward modeled in an explicit sense. Perhaps the authors can be more explicit about what physics are accounted for in the hazard forecast model.

Methods

Line 573: S_s is a derived parameter that combines porosity, bulk rock compressibility, water compressibility, and water density. Readers would benefit from knowing the parameters underlying the S_s values used in the study, particularly in the context of implementing or reproducing the model with a different simulation code. Specifically, there is no explicit mention of porosity – even though porosity can be backed out of S_s , it's somewhat standard to present porosity as part of a groundwater modeling project.

Lines 578 – 579: Assuming constant density & viscosity is equivalent to assuming isothermal conditions. What density & viscosity that are being assumed? This detail is highly relevant for this study because temperature-dependent viscosity varies over an order of magnitude between shallow and seismogenic depths. This variation would certainly affect pressure diffusion over the time scales being modeled here, albeit the magnitude of which is not certain without testing it numerically. If this cannot be tested with the model, then a statement acknowledging the limitation is warranted.

Line 581: Is 2.1 km the top of the model domain?

Line 584: "...depth of 20 km." Cranganu et al. (1998, Fig. 2) report a regional heat flux of ~ 50 mW/m² in central Oklahoma with a corresponding thermal gradient of $\sim 39^\circ\text{C}/\text{km}$. Over the 18 km thickness of this model, the resulting temperature distribution would range $\sim 700^\circ\text{C}$ (!). While it is understandable to set a non-realistically deep the bottom boundary to mitigate the potential for basal pressure feedbacks, I think the decisions underlying model geometry should include a broader discussion of model assumptions and the potential implications of these assumptions.

Cranganu, C., Lee, Y. and Deming, D., 1998. Heat flow in Oklahoma and the south central United States. *Journal of Geophysical Research: Solid Earth*, 103(B11), pp.27107-27121.

Lines 585 – 586: “We explicitly calculate all pressure changes in the model as relative to a uniform, pre-injection baseline.” Is this the initial conditions for the model? If so, the reader would benefit from knowing this baseline. How is it determined? Is it really uniform, i.e., the same from bottom of base of Arbuckle to 20 km? Wouldn’t a hydrostatic gradient be more appropriate, even for a coarse representation? If subject sentence does not reflect initial conditions, then they should be presented.

Lines 588 – 589: How far from the domain of interest are the model boundaries? What types of boundary conditions are imposed at the upper, lower and lateral boundaries? Dirichlet? Adiabatic? These are important details if the study is to be useable by the research & regulatory communities.

Lines 581 – 592: The model geometry is not clear. Is the model simply a rectangular volume with the Arbuckle as the uppermost layer of grid blocks? Although this is a fine assumption for such a large-scale model, the reader would benefit from a more transparent discussion about the assumptions that go into this geometry, particularly because the Arbuckle deepens and thickens substantially from northeast to southwest over the study area. The Texas Bureau of Economic Geology has some excellent maps illustrating these phenomena:

Arbuckle top: <http://www.beg.utexas.edu/index.php/gccc/co2-data/arbuckle-01>

Arbuckle thickness: <http://www.beg.utexas.edu/gccc/co2-data/arbuckle-03>

Lines 590 – 592: Discretization in the z-direction is not clear. Are the 12 layers of uniform thickness? If so, this would mean that each basement grid cell is 1500 m thick, so the vertical discretization would be 3× the horizontal discretization. Or do the layers have differing thickness, perhaps coarsening with depth? This should be clarified. Also, is MODFLOW stable for such large aspect ratios? Many finite difference schemes have problems when grid cell aspect ratio exceeds 2:1.

Line 607 – 612: Is the Nemaha fault an actual boundary condition or simply a region modeled with local permeability heterogeneity? The section reads as though the latter is true, but the authors describe the Nemaha as a “boundary condition.”

→ Based on the above comments, it seems that a supplemental figure illustrating the grid and initial/boundary conditions would be a tremendous help to the readers.

Extended Table 1: Add S_s for each lithology & model run, and perhaps the corresponding porosity.

Reviewer #3 (Remarks to the Author):

The manuscript "Physics-based forecasting of man-made earthquake hazards in Oklahoma and Kansas" by Langenbruch, Weingarten, and Zoback presents a method for developing probability forecasts for moderate or larger earthquakes in Oklahoma and Kansas due to saltwater disposal. The methodology depends on injection induced pressure and susceptibility of a local area to trigger, so called seismogenic index. The authors provide forecasts for seismicity rates through 2020 and conclude the method provides an improved way to forecast hazards from induced seismicity resulting from subsurface pressure increases.

The need for a physics-based approach to estimate seismic hazard from wastewater injection (or other anthropogenic activities) has been recognized for a few years and this manuscript is one of several new efforts to move this science forward. Through the applied methodology the authors achieve a significant improvement (at least in 2017, as shown in their fig 4) over a seismicity-only based methodology that has been used to date to generate the widely recognized USGS one-year hazard forecast (Peterson et al., 2016, 2017).

While the method appears promising, there is another recently published manuscript (Norbeck and Rubinstein, 2018) that also uses a physics-based approach based on fluid pressurization rates (and rate and state friction) to predict future seismicity. Norbeck and Rubinstein (2018), henceforth NR2018, and this manuscript, henceforth LWZ, start from a similar basis, e.g. that seismicity is governed by subsurface pressure changes and use the pressure evolution in the subsurface to forecast hazard. There are differences in the specific approaches employed, including the exact method that is used link pressures to seismicity changes and the inclusion of a spatially-variable seismicity index (SI) in the LWZ manuscript.

It is difficult to quantitatively judge the efficacy of the two methods because a direct comparison is not provided (through no fault of the authors here as NR2018 was published around the same time as LWZ was likely submitted). In my reading of the two articles I noted a some differences in the results that may be instructive for assessing the novelty of the LWZ methodology.

In particular, the NR2018 publication appears to be able to predict the onset of seismicity increase, I don't find any similar statement in the LWZ manuscript. Onset of significant seismicity is clearly an important trait of induced seismicity sequences to predict. LWZ forecasts seem to be made for time periods after end-2014 (close to the peak of seismicity) and the forecast (or maybe postdiction?) prior to ~2013/2014 does not fit particularly well to seismicity. It is not clear why the LWZ model does a relatively poor job predicting seismicity during the time period prior to ~2014 as it is not discussed in the manuscript. My assumption is the poor fit is because the SI background value has to be used prior to 2014 because the variable SI can only be determined when pressure increases in a month are above a given threshold (800 Pa/month) and if at least a minimum number of earthquakes (at least 5 M3+) occur within a 20 km cell. The authors should provide a map showing the dates that a variable SI can start to be computed in each cell.

The limitations on how quickly this method can be more reliably applied based on what is presented in the manuscript, it is not clear the statement in the conclusions regarding use of the LWZ method in other locations to predict seismicity is well-supported – my reading is that many years of relatively high seismicity are required for this method to work well. NR2018 appears to produce quite good forecasts without needing to appeal to spatial variability (that can only be assessed after significant seismicity has occurred), so it is not clear the LWZ method provides a significant improvement.

The manuscript does provide a discussion of the forecasted seismicity through 2020 that I found to be clearly presented.

Other comments:

Line 165: It is not clear how exactly 800 Pa per month is chosen as the cut off value.

Lines 236-238: This statement about the occurrence of $M \geq 3$ earthquakes does not seem relevant. The forecast shown in Fig. 3 is for $M4+$ earthquakes, so it is not clear why $M3+$ earthquakes should be used to indicate the model is performing well?

Line 241-242: It would be instructive to show the pressures evolution at the various depths in the model at one or two locations as supplemental figures.

Line 245-246: The authors note there is a disproportionally low probability for earthquake triggering for slow pressure increases – is there some physics to bear on the question of why this is?

Line 256-257: The paper seems to jump back and forth between forecasting $M3+$ and $M4+$, I understand the model can be used to forecast any magnitude threshold, but it would improve readability to remain consistent within the main figure texts and present the other magnitude forecasts in the supplemental.

Line 600&604: Provide a reference for the basement permeability and specific storage values used.

Line 611-12: Provide a reference for the idea that the Nemaha Fault is a structural trap

Line 673-5: It is not clear what is meant by this statement.

Line 679-82: The method seems to be applied inconsistently by manually setting SI in certain regions.

Reviewers' comments:

Reviewer #1:

Scope of Review: My expertise is in the field of nonisothermal, multi-phase fluid flow in porous and fracture geologic media with a specific emphasis on modeling subsurface fluid injections. Consequently, the scope of this review is limited to the fluid aspects of fluid-triggered earthquakes. I am not qualified to provide rigorous peer-review in the area of seismological processes. General Comments: This manuscript presents a novel, physics-based method to estimate spatiotemporal variations of fluid-triggered earthquakes using north-central Oklahoma and southern Kansas as the study area for development and testing. In this study, the authors modify the Seismogenic Index calculation to account for deep underground injections of oilfield wastewater, and the resulting fluid pressure changes that are known to trigger earthquakes. The novel approach taken in this study is designed to ameliorate the problem of non-stationarity in calculations of fluid-triggered earthquake hazard, which traditionally assumes a constant earthquake rate. Overall this is a timely and provocative idea, and one that is ideally suited for rapid publication in a widely read scientific journal, such as Nature-Communications. While I do support publishing this study, I think there is also room for improvement, particularly in the context of how the numerical groundwater model is presented to the readership. Many of the specific comments below are based on the idea that the methods must be clearly articulated in order for this method to be usable by the larger community working on fluid-triggered earthquakes. For example, a number of comments relate to more clearly presenting the model geometry, initial/boundary conditions, and presenting a very clear discussion of the model assumptions, as well as the potential implications of these various assumptions on the model results. Given the massive size and scope of the groundwater modeling presented in this study, these comments should not be perceived as indication that the modeling is inadequate—in contrast, the regional scale groundwater model is an impressive accomplishment—however, I do believe that it is critically important for the reader to understand the model assumptions & limitations. The other specific comment that is worth mentioning here is that I had some difficulty understanding the concept of “pressure rate” as it is presented in the manuscript & supplemental. I think additional discussion would be very useful to the reader in making sense of this important concept. In conclusion, this is an important study and I support publication after modest revisions to the groundwater modeling discussion & methods.

Signed,

Ryan M. Pollyea

Specific Comments

2) Line 74: spatially variability => spatial variability

Done

3) Lines 107 – 120: Calibrating the permeability against earthquake is an interesting approach.

Was permeability anisotropy considered? Or porosity (through S_s)?

The main goal of the hydrogeologic model effort was to quantify the permeability heterogeneity between the injection formation (the Arbuckle Group) and the crystalline basement. This permeability contrast is physically related to the spatiotemporal delay between where active injection is occurring and earthquake occurrence. Therefore, all of our model sensitivity analysis focused on the interplay between the permeability of these two formations. Neither permeability anisotropy nor heterogeneity of porosity were considered and are beyond the scope of this study.

4) Line 114 – 115 & Fi. 1b: This study relies heavily on the concept of “pressure rate,” which is defined in Extended Fig. 5a as, dP_f/dt . After reading the primary manuscript, supplemental material, and references 28-29, I can’t seem to figure out what “pressure rate” actually means in the context of the paper. Figure 1b shows that the pressure rate at 6.5 km depth reaches ~ 25 MPa/month in 2016 – this is a lot of fluid pressure, and perhaps a bit misleading because Fig 1c shows ~ 200 kPa of pressure accumulation across the model domain after 17 years of injection. How does 25 MPa per month only result in 200 kPa of pressure after 17 years? Is the pressure data in Fig 1b the total pressure accumulation over the complete set of seed points per time? If so, I’m not sure what this value actually means because the seed points are used as a proxy for potential fault locations, so the cumulative pressure across all seed points is less important than the local pressure accumulation at each one. Would average ΔP as a fn of time serve as a better proxy for changes to effective stress at 6.5 km depth? This would more clearly connect changes in the overall pressure regime to the faults (seed points) that are potentially affected, while also allowing a measure of uncertainty about the pressure curve to be shown. Regardless of how the authors choose to represent the overall pressure change at seismogenic depths, I think that a more information is needed in the supplemental to clearly articulate the concept of pressure rate.

We understand that the way the pressure rate concept was presented in the originally submitted manuscript (in Fig. 1) was a bit misleading. In the original Figure 1b, we presented the monthly pressure increase summed up over all seed points. In the revised Figure 1b we present the average monthly pressure increase at 6.5km depth, which allows a direct comparison to the pressure increase shown on the map in Fig. 1 c and d. Thank you for your comment! We also included more details on the concept of pressure rate in the text. It is correct that the local pressure accumulation at each seed point is more important than the summed up rate at all seed points. Fig. 1 gives a large-scale overview about the relation between saltwater injection, induced earthquakes and pressure increase. Our SI model of course considers the monthly pressure accumulation at each individual seed point to produce the seismic hazard maps.

Revised and added text in the manuscript:

(Lines 110-113): “Following the concept that changes in earthquake activity are caused by changes of the stressing (pressure) rate^{29,30}, we vary the basement permeability in our model and compare the rate of pressure increase (averaged over all points) to the overall observed earthquake rate ($M \geq 3$).”

(Lines 114-116): “Pressure rate (and rate of pressure increase) in this publication always refers to monthly pressure accumulation at a given point in space.”

Lines (150-161): “We set up a model to transfer injection-induced pressure increases in time and space into seismicity rates by reconstructing how triggering probabilities of observed earthquakes are related to modelled pressure rates. High pressure rates (fast pressure increases) are driving faults faster towards failure by reducing the effective normal stress. Compared to low pressure rates more faults are expected to reach the failure stress in a given time and we expect faster pressure increases to be more likely associated with earthquakes. The relation between pressure rate and earthquake probability (Fig. 2a) reconstructed from our model (see Methods for details) confirms the expected increase of earthquake probabilities with the rate of pressure increase. Earthquake probabilities in Oklahoma and Kansas are increasing with the square of the pressure rate (Fig. 2a). Slow pressure increase is causing a disproportionately low percentage of seismicity.”

5) Figure 1b: In this figure, it is not clear what data are observed and what data are modeled. It seems reasonable that volume and earthquake data are observed/measured through 2017, and pressure data is modeled; however, this should probably be stated in the caption – particularly, because everything ≥ 2018 is simulated.

We clarify which data is modelled or observed in the caption of the figure in the revised manuscript. See caption of revised Fig. 1.

6) Figures 1b & 1c: These panels use the same color scheme, but different range – maybe mention the different range in the caption.

We mention the different color ranges in the caption of the figure in the revised manuscript. See revised caption of Fig. 1.

7) Line 212: Typo: opne => one

Done

8) Lines 293 – 295: It is clear that the hazard model developed here is driven by spatial and temporal variations in pressure accumulation at seismogenic depths, but it is less clear that the model takes into account the spatial variability of optimally aligned faults b/c the overall method incorporates pressure accumulation rate into the SI calculation, which is dependent on earthquake rate. I don't see where a priori knowledge about fault location/geometry is part of the model, except perhaps in the context of earthquake occurrence, but this can only be forward modeled in a probabilistic sense, whereas pressure diffusion in this study is forward modeled in an explicit sense. Perhaps the authors can be more explicit about what physics are accounted for in the hazard forecast model.

In the revised manuscript, we clearly state that our model is a hybrid physical-statistical approach. A priori knowledge about fault location/geometry is not part of the model. The SI represents local number and stress state of pre-existing faults affected by the pressure increase in a statistical sense. We clarify this point in the revised manuscript and give more explanations about why the SI represents number and stress state of faults.

Added and revised text in the manuscript:

(Lines 10-13): "In our hybrid physical-statistical model induced seismicity is driven by the rate of injection-induced pressure increases at depth at any given location and spatial variations in the number and stress state of pre-existing basement faults affected by the pressure increase."

(Lines 171-175): "The quantity $10^{\Sigma_p(\bar{r})}$ in Eq. 1 is a proxy for number and stress state of basement faults at location \bar{r} . The higher the SI at a location the higher the earthquake rate caused by a given pressure increase, because a higher number of (or more critically stressed) pre-existing faults are affected by the pressure increase."

(Lines 196-203): "We find that $\Sigma_p(\bar{r})$ varies by about two units across the study area (Fig. 2 b-f). This would mean that in areas of highest SI, a given pressure increase will cause one hundred times as many earthquakes as in areas characterized by the lowest SI. The SI maps (Fig. 2 b-f) can be interpreted as a map of critically stressed faults in the crystalline basement in the areas affected by pressure changes. The areas of highest SI imply one hundred times as many pre-existing faults than the areas of lowest SI. Thus, both the local pressure increase and the local seismogenic state must be combined to determine the local rate of induced earthquakes."

Methods

9) Line 573: Ss is a derived parameter that combines porosity, bulk rock compressibility, water compressibility, and water density. Readers would benefit from knowing the parameters underlying the Ss values used in the study, particularly in the context of implementing or reproducing the model with a different simulation code. Specifically, there is no explicit mention of porosity – even though porosity can be backed out of Ss, it's somewhat standard to present porosity as part of a groundwater modeling project.

We have added a supplemental table (new Supplementary Table 1) to the manuscript which describes the porosity, bulk rock compressibility, bulk water compressibility and water density used in the calculation of specific storage for

the Arbuckle Group and crystalline basement units, respectively. The values of all material property inputs are derived from the range of reported values in the literature.

We have also added additional text to address the uncertainty in how S_s was derived:

(Lines 635-648): “Specific storage in our model is a parameter derived from the porosity, bulk compressibility of water and bulk compressibility of rock reported in the literature for the Arbuckle Group and crystalline basement (Supplementary Table 1). Specific storage combines the bulk compressibility of water (B_w), bulk compressibility of rock (B_r), fluid density (ρ) and porosity (ϕ) as described below:

$$S_s = \rho_w g (\beta_w \phi + \beta_r)$$

We use a fluid density of 1062 kg/m³ reflective of brine with TDS of 100,000 ppm [Nelson et al. (2015)] at a reservoir temperature of 50 deg C. The bulk compressibility of water of $4.4 \times 10^{-10} \text{ Pa}^{-1}$ respects the reservoir temperature. The bulk compressibility and porosity of the Arbuckle Group is $0.16 \times 10^{-10} \text{ Pa}^{-1}$ and 0.20 derived from direct observations and core testing [Kroll et al. (2017); Perilla (2017); Carr et al. (1986)]. The bulk compressibility and porosity of the crystalline basement is set from literature value to be $0.7 \times 10^{-11} \text{ Pa}^{-1}$ and 0.01, respectively [Freeze and Cherry et al. (1979)].”

We have added additional references for the supplemental table:

Kroll, K. A., Cochran, E. S., & Murray, K. E. (2017), Poroelastic properties of the Arbuckle Group in Oklahoma derived from well fluid level response to the 3 September 2016 M w 5.8 Pawnee and 7 November 2016 M w 5.0 Cushing earthquakes. *Seismological Research Letters*, 88(4), 963-970.

10) Lines 578 – 579: Assuming constant density & viscosity is equivalent to assuming isothermal conditions. What density & viscosity that are being assumed? This detail is highly relevant for this study because temperature-dependent viscosity varies over an order of magnitude between shallow and seismogenic depths. This variation would certainly affect pressure diffusion over the time scales being modeled here, albeit the magnitude of which is not certain without testing it numerically. If this cannot be tested with the model, then a statement acknowledging the limitation is warranted.

We have added text describing the density and viscosity used in the modeling as well as a supplemental table of material properties (new Supplementary Table 1) describing the source of the parameters.

Added text in the revised manuscript (Lines 600-604): “The fluid density and fluid dynamic viscosity in the model are 1062 kg/m³ and 0.547 cP, respectively (Supplementary Table 1). To calculate the fluid density and dynamic viscosity used in the model, we assume a brine TDS of 100,000 ppm⁴² and a reservoir temperature of 50 °C derived from a standard geothermal gradient of 25 °C per km.”

11) Line 581: Is 2.1 km the top of the model domain?

Yes. We added text to describe why the top of the model domain is 2.1 km:

Lines 616-617: “The upper boundary is reflective of the confined condition of the Arbuckle Group reservoir [Keranen et al. (2014)].”

12) Line 584: “...depth of 20 km.” Cranganu et al. (1998, Fig. 2) report a regional heat flux of ~50 mW/m² in central Oklahoma with a corresponding thermal gradient of ~39°C/km. Over the 18 km thickness of this model, the resulting temperature distribution would range ~700°C(!). While it is understandable to set a non-realistically deep the bottom boundary to mitigate the potential for basal pressure feedbacks, I think the decisions underlying model geometry should include a broader discussion of model assumptions and the potential implications of these assumptions.

Cranganu, C., Lee, Y. and Deming, D., 1998. Heat flow in Oklahoma and the south central United States. *Journal of Geophysical Research: Solid Earth*, 103(B11), pp.27107-27121.

Yes, we set the bottom boundary to a very deep depth to mitigate the first-order boundary effect of a no-flow boundary at the model base. While we understand that 20 km depth is non-realistic at some level, our primary concern was to not have the bottom boundary effect the pressure calculations at the depth of seismicity. We have added a sentence in Lines 618-619: The lower boundary of the model domain is set intentionally very deep so as to have no effect on pressure calculations at hypocentral depths.

13) Lines 585 – 586: “We explicitly calculate all pressure changes in the model as relative to a uniform, pre-injection baseline.” Is this the initial conditions for the model? If so, the reader would benefit from knowing this baseline. How is it determined? Is it really uniform, i.e., the same from bottom of base of Arbuckle to 20 km? Wouldn't a hydrostatic gradient be more appropriate, even for a coarse representation? If subject sentence does not reflect initial conditions, then they should be presented.

We have updated the text to reflect that the initial conditions of the model are calculated as pressure change relative to uniform, hydrostatic pre-injection baseline.

14) Lines 588 – 589: How far from the domain of interest are the model boundaries? What types of boundary conditions are imposed at the upper, lower and lateral boundaries? Dirichlet? Adiabatic? These are important details if the study is to be useable by the research & regulatory communities.

We have added a supplementary figure (Supplementary Figure 13) and text describing the model domain and modeled boundaries. All modeled boundaries are of Neumann Type (no-flow). The upper boundary (above the Arbuckle) reflects the confined condition of the Arbuckle Group aquifer. The lower and lateral boundaries are set far from injection wells to have no effect on pressure calculations.

Lines 614-616: “All model boundaries are of Neumann Type (no-flow). Model boundaries in x and y were set intentionally far from the dominant region of injection such that they have no effect on pressure calculations. The upper boundary is reflective of the confined condition of the Arbuckle Group reservoir. The lower boundary is set intentionally very deep to have no effect on pressure calculations.”

15) Lines 581 – 592: The model geometry is not clear. Is the model simply a rectangular volume with the Arbuckle as the uppermost layer of grid blocks? Although this is a fine assumption for such a large-scale model, the reader would benefit from a more transparent discussion about the assumptions that go into this geometry, particularly because the Arbuckle deepens and thickens substantially from northeast to southwest over the study area. The Texas Bureau of Economic Geology has some excellent maps illustrating these phenomena:

Arbuckle top: <http://www.beg.utexas.edu/index.php/gcccc/co2-data/arbuckle-01>

Arbuckle thickness: <http://www.beg.utexas.edu/gcccc/co2-data/arbuckle-03>

We have added a supplementary figure (Supplementary Figure 13) describing the model domain, layered structure and permeability heterogeneity of the model. The model is a rectangular volume with the Arbuckle Group represented by the uppermost layer of grid blocks. We use the average thickness of the Arbuckle Group across the entire domain.

16) Lines 590 – 592: Discretization in the z-direction is not clear. Are the 12 layers of uniform thickness? If so, this would mean that each basement grid cell is 1500 m thick, so the vertical discretization would be 3× the horizontal discretization. Or do the layers have differing thickness, perhaps coarsening with depth? This should be clarified.

Also, is MODFLOW stable for such large aspect ratios? Many finite difference schemes have problems when grid cell aspect ratio exceeds 2:1.

We have added a supplementary figure describing the model domain (Supplementary Figure 13) which also shows the vertical discretization of the model. MODFLOW is stable for aspect ratios up to 10:1 and the maximum aspect ratio used in our model is ~2.5:1.

17) Line 607 – 612: Is the Nemaha fault an actual boundary condition or simply a region modeled with local permeability heterogeneity? The section reads as though the latter is true, but the authors describe the Nemaha as a “boundary condition.”

We have added text to clarify the misconception that the Nemaha Fault is simply a low-permeability barrier to flow and not a boundary condition.

Lines 650-652: “Beyond the layered heterogeneity represented in our model, a key feature in our model is the representation of the Nemaha Fault as a regional-scale, low-permeability barrier to cross-fault flow ($k = 10^{-20} \text{ m}^2$).”

18) => Based on the above comments, it seems that a supplemental figure illustrating the grid and initial/boundary conditions would be a tremendous help to the readers.

We have added a supplementary figure showing the model domain (Supplementary Figure 13), parameterization of the model units, the boundary conditions and vertical grid discretization.

19) Extended Table 1: Add Ss for each lithology & model run, and perhaps the corresponding porosity.

We have added a supplementary table including the porosity, fluid density, fluid viscosity, bulk compressibility of water, bulk compressibility of rock and specific storage for each model unit (Supplementary Table 1).

Reviewer #3 (Remarks to the Author):

The manuscript “Physics-based forecasting of man-made earthquake hazards in Oklahoma and Kansas’ by Langenbruch, Weingarten, and Zoback presents a method for developing probability forecasts for moderate or larger earthquakes in Oklahoma and Kansas due to saltwater disposal. The methodology depends on injection induced pressure and susceptibility of a local area to trigger, so called seismogenic index. The authors provide forecasts for seismicity rates through 2020 and conclude the method provides an improved way to forecast hazards from induced seismicity resulting from subsurface pressure increases.

20) The need for a physics-based approach to estimate seismic hazard from wastewater injection (or other anthropogenic activities) has been recognized for a few years and this manuscript is one of several new efforts to move this science forward. Through the applied methodology the authors achieve a significant improvement (at least in 2017, as shown in their fig 4) over a seismicity-only based methodology that has been used to date to generate the widely recognized USGS one-year hazard forecast (Petersen et al., 2016, 2017).

Note that our physics-based approach also presents an improved way to forecast hazards from induced seismicity, because of the two following points:

- 1. Our method presents a significant advance, because it can be used to identify the optimal injection strategy to mitigate the remaining seismic hazard. Added text in the revised manuscript (Lines 258-265): “More importantly, a significant advance of our physics-based method is that it can be used to identify the optimal injection strategy to mitigate the remaining seismic hazard. Alternative scenarios for future injection rates in space and time could easily be considered to evaluate how possible injection regulations*

would affect the seismic hazard. Not only further reduction in total injection volume, but also a redistribution of injection volumes in the study area could mitigate the seismic hazard. Moving injection away from critical regions of high SI could lower the induced seismic hazard without reducing the overall volume of injection.”

2. *Our physics-based method is not restricted to short term (1-year) forecasts (see revised Figure 4a). Added text in the revised manuscript (Lines 288-292): “Fig. 4a demonstrates that changing the temporal endpoint of the calibration between Dec 2011 and Dec 2017 has no significant effect on forecasted seismicity rates. Had we only used earthquake information through Dec 2011 to calibrate the SI, our model would have successfully predicted the increase, peak and decrease of the large-scale seismicity rate in response to changes of injection rates.”*

21) While the method appears promising, there is another recently published manuscript (Norbeck and Rubinstein, 2018) that also uses a physics-based approach based on fluid pressurization rates (and rate and state friction) to predict future seismicity. Norbeck and Rubinstein (2018), henceforth NR2018, and this manuscript, henceforth LWZ, start from a similar basis, e.g. that seismicity is governed by subsurface pressure changes and use the pressure evolution in the subsurface to forecast hazard. There are differences in the specific approaches employed, including the exact method that is used link pressures to seismicity changes and the inclusion of a spatially-variable seismicity index (SI) in the LWZ manuscript.

There are significant differences and advances in all components of our model.

1. *The incorporation of a spatially variable seismogenic state of the crust (the SI) is one of the major advances in our paper. We separate two conditioned, which must be fulfilled to trigger seismicity. Pressure must increase and there must be critically stressed pre-existing faults. Once the SI, a proxy for number and stress state of faults affected by the pressure increase, can be calibrated on the local scale, local-scale seismicity caused by future pressure increase can be forecasted.*
2. *In the revised manuscript, we evidently demonstrate that local-scale model calibration is required to forecast local scale seismicity (new Supplementary Figure 5). Results and applicability of NR2018 are overstated in the original publication. We find that, in most parts of the study area, their model cannot be applied on the local scale (new Supplementary Figure 5). On the large scale (complete study area) NR2018 is significantly under-predicting the decay of seismicity (see revised Fig. 4a).*
3. *Our model considers real locations and injection rates of all injection wells in the study area. We developed a full physics-based hydrogeologic model and consider pressure migration into the basement where the seismicity is occurring. In contrast the NR2018 model considers the Arbuckle group as a closed system (on all scales of interest), no pressure migration into surrounding formations (including the crystalline basement where seismicity occurs) is considered. Equally distributed injection across the region is assumed. Real locations and rates of individual wells are not considered.*
4. *Because of the closed system assumption, pressure rates in NR2018 are directly proportional to summed up injection rates in all considered study areas. As we present in our new Supplementary Figure 4, temporal evolution and peak of seismicity rates in some local-scale study regions are completely unrelated to injection rates. Considering the Arbuckle group as an open system is essential to relate injection rates, pressure increase and seismicity.*
5. *Our model forecasts maps of occurrence probabilities of potentially damaging magnitudes (Fig. 3). The maps can be used to identify the optimal injection strategy to mitigate the remaining seismic hazard. NR2018 only presents a method to model M3+ seismicity rates.*

Please note that we did not include all detailed about the comparison to NR2018 (see replies to comments 21-25) in the revised manuscript, because it is important to point out that our method stands on its own, and

represents a significant advance in the complete field of seismic hazard forecasting and physical understanding of induced earthquakes.

22) It is difficult to quantitatively judge the efficacy of the two methods because a direct comparison is not provided (through no fault of the authors here as NR2018 was published around the same time as LWZ was likely submitted). In my reading of the two articles I noted some differences in the results that may be instructive for assessing the novelty of the LWZ methodology.

In the revised manuscript, we present a direct comparison of our model and the NR2018 model. The results clearly demonstrate that our method presents a significant advance in physical justification, hydrogeologic model design and forecasting performance (revised Fig. 4a and new Supplementary Figures 4 and 5).

Added text in the revised manuscript (Lines 305-311): "In Fig. 4a and Supplementary Figure 5 we compare our results to a recent paper³⁷ which also considers injection rates to model seismicity rates in Oklahoma and Kansas based on rate and state friction³⁸. It considers the Arbuckle group as a closed hydrogeologic system with equally-distributed injection rates throughout the region. Because no forecast of the seismicity rate (2018-2020) is presented in the publication, we forecast the seismicity rate through Dec 2020 based on the rate-and-state model³⁷ assuming constant injection rates after March 2018 as in our analysis.

While the large-scale seismicity rates resulting from the rate-state and the SI models are similar from 2012 through 2015, the rate-state model significantly under-predicts the decay rate following reduced injection rates (Fig. 4a). Moreover, as the rate-and-state model does not consider spatial variability of model parameters, it is not able to describe onset, increase, peak or decay of seismicity in any of the six local-scale regions presented in Supplementary Figure 5."

(See also replies to comments 20-25).

23) In particular, the NR2018 publication appears to be able to predict the onset of seismicity increase, I don't find any similar statement in the LWZ manuscript. Onset of significant seismicity is clearly an important trait of induced seismicity sequences to predict. LWZ forecasts seem to be made for time periods after end-2014 (close to the peak of seismicity) and the forecast (or maybe postdiction?) prior to ~2013/2014 does not fit particularly well to seismicity. It is not clear why the LWZ model does a relatively poor job predicting seismicity during the time period prior to ~2014 as it is not discussed in the manuscript. My assumption is the poor fit is because the SI background value has to be used prior to 2014 because the variable SI can only be determined when pressure increases in a month are above a given threshold (800 Pa/month) and if at least a minimum number of earthquakes (at least 5 M3+) occur within a 20 km cell. The authors should provide a map showing the dates that a variable SI can start to be computed in each cell.

On the large-scale (see revised Fig. 4a) it appears as if NR2018 predicts onset and early phase of seismicity more accurately than the SI model. However, what seems advantageous results from the unjustified assumption of an equally distributed injection rate across the complete study area. Real locations and injection rates of individual wells are not considered in the model. More importantly, the model proposed by Norbeck and Rubinstein does not consider spatial variability of model parameters and is not able to describe onset, increase, peak or decay of seismicity in any of the six local-scale regions presented in Supplementary Figure 5.

Because all existing forecasting methods need earthquake data to calibrate model parameters, the onset of seismicity is always a postdiction. The onset of the large-scale seismicity, described by the NR2018 model, is based on tuning the model parameters using the complete earthquake catalog through December 2017 (see also reply to comment 24).

We revised our method (Eq. 1) to be able to use a variable SI as soon as possible. The revised method does not consider different SI values for slow and fast pressure increase (see also reply to comments 26). Using the revised method improved the fit to seismicity prior to 2013/2014. SI models calibrated through early times (Dec 2011- Dec 2013) show a reasonable fit to the early phase of seismicity (see revised Fig. 4a). Models calibrated through later

times (Dec 2014- Dec 2017) still tend to overestimate the early phase of seismicity, but observed earthquakes rate fall within the uncertainty range of our model.

In revised Fig. 2 b-e we present SI maps calibrated through different times to illustrate the dates that a variable SI can be computed using the revised method. Using earthquakes through Dec 2014 a variable SI can be computed in most parts of the region of seismic concern and we forecast hazard maps for 2015-2020. Note that on the large scale our method can be used as early as Dec 2011 (revised Fig. 4a).

24) The limitations on how quickly this method can be more reliably applied based on what is presented in the manuscript, it is not clear the statement in the conclusions regarding use of the LWZ method in other locations to predict seismicity is well-supported – my reading is that many years of relatively high seismicity are required for this method to work well. NR2018 appears to produce quite good forecasts without needing to appeal to spatial variability (that can only be assessed after significant seismicity has occurred), so it is not clear the LWZ method provides a significant improvement.

We removed the statement about the applicability to other regions, because we don't want to overstate our results. Application to other regions would require recalibration of the SI and adjustment of the hydrogeologic model to local geology in the new study area.

We understand that reading the publication by Norbeck and Rubinstein gives the impression, that their model can be applied without appealing to heterogeneity. However, results and applicability of NR2018 seem to be overstated in the NR2018 publication.

- 1) *We find that, in most parts of the study area, the NR2018 method cannot be successfully applied to local-scale regions (new Supplementary Figure 5 in the revised manuscript).*

Added text in the revised manuscript (Lines 314-317): "Moreover, as the rate-and-state model does not consider spatial variability of model parameters, it is not able to describe onset, increase, peak or decay of seismicity in any of the six local-scale regions presented in Supplementary Figure 5."

- 2) *We evidently demonstrate that, because our model is considering spatial heterogeneity it forecasts local-scale seismicity rates more successfully (new Supplementary Figure 5).*

Added text in the revised manuscript (Lines 293-301): "Ultimately, the most important application of our model is that it can be used to predict local-scale seismicity rates, because the SI is calibrated on the local-scale. In Supplementary Figure 5 we present seismicity rate forecasts in six local-scale study areas of 25 km radius. The areas are selected to represent the full value range of the SI identified across the complete study area (Supplementary Figure 6). Our results show that our model can be used to forecasts local-scale seismicity rates. However, because the local-scale SI can vary significantly from the large-scale average, local-scale model calibration (based on local seismicity and pressure increase) is required to successfully forecast local-scale seismicity rates."

- 3) *It is not true that NR2018 produce predictions based on a smaller number of past earthquakes. The complete earthquake catalog through Dec 2017 is used to "tune" the parameters. As shown in NR2018 Supplementary Fig. S8 seismicity rates resulting from "untuned" parameters do not fit the observations. See also the following text (S4. Seismicity rate forecast in the Supplementary material of NR2018): "In Eq. 1 (in NR2018), there are two main groups of parameters that affect seismicity rate evolution in different ways. The first is \dot{s} / \dot{s}_0 . This ratio controls the magnitude of the seismicity rate change. The second is $a\sigma / \dot{s}_0$. This term controls the variable time lag. To effectively match the observed behavior, both of these terms must be tuned to get both the timing and the magnitude of the seismicity rate correct."*

- 4) *It is correct, that local seismicity is needed for local-scale calibration of our model (new Supplementary Figure 5 in the revised manuscript). Note, however, that on the large scale our model can be used as early as Dec 2011 (revised Figure 4a).*

Added text in the revised manuscript (Lines 288-292): “Fig. 4a demonstrates that changing the temporal endpoint of the calibration between Dec 2011 and Dec 2017 has no significant effect on forecasted seismicity rates. Had we only used earthquake information through Dec 2011 to calibrate the SI, our model would have successfully predicted the increase, peak and decrease of the large-scale seismicity rate in response to changes of injection rates.”

- 25) The manuscript does provide a discussion of the forecasted seismicity through 2020 that I found to be clearly presented.

Thank you for your comment. Yes, we present real forecasts based on future injection rates.

Because NR2018 do not present a forecast through 2020, we use their model and the assumption of a constant rate of injection after March 2018 (as in our model) to produce a forecast through 2020. The results shown in Fig. 4a demonstrate that, in contrast to our model, the NR2018 model significantly under-predicts the decline of the seismicity rate following reduced injection rates.

Added text in the revised manuscripts (Lines 308-311): “Because no forecast of the seismicity rate (2018-2020) is presented in the publication, we forecast the seismicity rate through Dec 2020 based on the rate-and-state model assuming constant injection rates after March 2018 as in our analysis.

While the large-scale seismicity rates resulting from the rate-state and the SI models are similar from 2012 through 2015, the rate-state model significantly under-predicts the decay rate following reduced injection rates (Fig. 4a). Moreover, as the rate-and-state model does not consider spatial variability of model parameters, it is not able to describe onset, increase, peak or decay of seismicity in any of the six local-scale regions presented in Supplementary Figure 5.”

Please also note that NR2018 do only present a method to forecast M3+ seismicity rates. Using our method, we forecast maps of occurrence probabilities of potentially damaging magnitudes (Fig. 3).

Other comments:

- 26) Line 165: It is not clear how exactly 800 Pa per month is chosen as the cut off value.

We revised our method to make the application more consistent. The revised method does not consider different SI values for slow and fast pressure increase and does not consider a cut off value (which can't be easily identified). We describe the over-proportional increase of earthquake probabilities with the pressure rate using a continuous function given by the square of the pressure rate and the SI (see revised Eq.1 and revised Fig. 2a).

- 27) Lines 236-238: This statement about the occurrence of $M \geq 3$ earthquakes does not seem relevant. The forecast shown in Fig. 3 is for M4+ earthquakes, so it is not clear why M3+ earthquakes should be used to indicate the model is performing well?

The statement was based on the fact that probabilities of M4+ in our model will be high where a high number of M3+ earthquakes is expected. However, because the hazard maps are developed to inform about potentially damaging earthquakes (M4+), we removed the statement in the revised manuscript and focus on the occurrence of M4+ earthquakes.

Revised text (Lines 237-241): “We find that earthquakes observed in the year of the predictions occur where our model forecasts enhanced exceedance probabilities (Fig. 3 a-d). 64 of 65 $M \geq 4$ earthquakes recorded from Jan 2015 through May 2018 occurred where our model predicts annual exceedance probabilities above 10%. 57 of 65 $M \geq 4$ earthquakes occurred within contours of 30% exceedance probability.”

28) Line 241-242: It would be instructive to show the pressures evolution at the various depths in the model at one or two locations as supplemental figures.

We have output the pressure evolution from the model at 5, 6, 6.5 and 7 km depth at X and Y locations to demonstrate the effect of the pressure evolution at depth in the model through time (see new Supplementary Figure 8). In general, increasing the depth of interest in the model delays the arrival of pressure diffusion as well as dampens it the pressure effect. Pressure increase is slowing down over a wide range of depths (see new Supplementary Figure 8).

Added text in the revised manuscript (Lines 242-243): “In response to decreased saltwater injection rates, pressure increases are slowing down over a wide range of depths (Supplementary Figure 8) and our model forecasts a wide-spread reduction of the seismic hazard in 2017.”

29) Line 245-246: The authors note there is a disproportionally low probability for earthquake triggering for slow pressure increases – is there some physics to bear on the question of why this is?

We added text in discussion section of the revised manuscript to explain possible physical processes explaining why we observe disproportionally low probabilities for slow pressure increase.

Added text in the discussion section of the revised manuscript (Lines 326-333): “Other studies observe a direct proportionality between pressure increase (injection rates) and earthquake probability during hydraulic stimulation of enhanced geothermal systems and hydraulic fracturing sites. However, we find that earthquake probabilities in Oklahoma and Kansas are increasing with the square of the pressure rate (Fig. 2a). Slow pressure increases cause a disproportionally low percentage of seismicity. Relatively more aseismic fault creep could be occurring when there are slow pressure increases. Moreover, aftershock triggering might contribute to the over-proportional increase of probabilities with the rate of pressure increase.

30) Line 256-257: The paper seems to jump back and forth between forecasting M3+ and M4+, I understand the model can be used to forecast any magnitude threshold, but it would improve readability to remain consistent within the main figure texts and present the other magnitude forecasts in the supplemental.

We understand that we didn't make clear enough why we discuss M3+ or M4+ in the different sections of the manuscript. Model calibration and performance evaluation should always use the highest number of observations to ensure statistical significance. Thus, we select a magnitude threshold of M=3 in some sections of the manuscript discussing calibration and performance of the model.

In other section, where the seismic hazard is discussed, we select a magnitude threshold of M=4 because smaller magnitudes are unlikely to cause damage.

Added and revised text in the revised manuscript (Lines 267-274) and new Supplementary Figure 9: “To retrospectively evaluate the performance of our model, we compare the observed seismicity rate ($M \geq 3$) in the complete study area to the forecasted seismicity rate resulting from SI models calibrated through different temporal endpoints. We select a magnitude threshold of M=3, because the model performance should be tested based on the highest possible number of observations. Note that while the observed rate of $M \geq 4$ earthquakes falls well within the uncertainty range of our model (Supplementary Figure 9), it does not allow us to draw statistically significant conclusions, because of the high uncertainty caused by the smaller number of $M \geq 4$ observations.”

31) Line 600&604: Provide a reference for the basement permeability and specific storage values used.

We have added one reference for the permeability of fractured crystalline basement rock to line 629-630 in the revised manuscript and cited references already in the text as well.

Added reference: Shapiro, S.A., Huenges, E., and Borm, G. (1997), Estimating the crust permeability from fluid-injection-induced seismic emission at the KTB site: Geophysical Journal International, v. 131, p. F15–F18.

We have added references for the specific storage values used in lines 646-648 of the revised manuscript:

- 1.) Domenico, P. A., & Schwartz, F. W. (1998). Physical and chemical hydrogeology (Vol. 506). New York: Wiley.*
- 2.) Freeze, R. A., & Cherry, J. A. (1979). Groundwater. In Groundwater. Prentice-Hall.*

32) Line 611-12: Provide a reference for the idea that the Nemaha Fault is a structural trap

We have added two references for the idea that the Nemaha Fault is a structural trap to lines 654-655.

- 1) Dolton, G. L., and T. M. Finn (1989), Petroleum geology of the Nemaha uplift, central Midcontinent: U.S. Geol. Surv. Open File Rep., 88–450D, p. 39*
- 2) McBee Jr, W. Nemaha strike-slip fault zone. AAPG Mid-Continent Section Meeting. 2003.*

33) Line 673-5: It is not clear what is meant by this statement.

We revised our method to make the application more consistent. The statement is removed, because the revised method does not consider different SI values for slow and fast pressure increase. We describe the over-proportional increase of earthquake probabilities using a continuous function given by the square of the pressure increase and a constant SI at all pressure rates.

34) Line 679-82: The method seems to be applied inconsistently by manually setting SI in certain regions.

We revised our method to make the application more consistent.

Added and revised text (Lines 719-727): “Where the number of earthquakes is not sufficient to directly calibrate the SI based on observed earthquakes and modelled pressure rates, we apply the following rule to obtain a SI value. If no directly calibrated SI value exist within 40 km radius around a selected seed point, the SI is set to the mean value of all directly calibrated SI values. If directly calibrated SI values exist within 40 km radius, the SI is interpolated (using the griddata function in Matlab). Dotted areas in Supplementary Figure 6 show areas where interpolated or averaged SI values are used to create the seismic hazard maps shown in Fig. 3.”

REVIEWERS' COMMENTS:

Reviewer #1 (Remarks to the Author):

Second Peer Review Comments for

Physics-based forecasting of man-made earthquake hazards in Kansas

By C. Langenbruch, M. Weingarten, and M. Zoback

Scope of Review: My expertise is in the field of nonisothermal, multi-phase fluid flow in porous and fractured geologic media with a specific emphasis on modeling subsurface fluid injections. Consequently, the scope of this review is limited to the *fluid* aspects of fluid-triggered earthquakes. I am not qualified to provide rigorous peer-review in the area of seismological processes.

General Comments: My initial review of the subject manuscript focused on several topics, including the concept of pressure rate (what it means and how it is calculated) and hydrogeological model presentation. In the revised manuscript, the authors adequately address all of my previous concerns, particularly in the context of the pressure rate concept. Specifically, I think the revised Figure 1 is much more accessible to the reader. The additional detail on model parameterization and corresponding rationale also enhances reproducibility and facilitates implementation by third parties.

I have also reviewed the discussion between the authors and Reviewer #3, which focuses on a comparison between the current study and the coupled hydro-mechanical model recently published by Norbeck and Rubenstein (2018). I was aware of this latter study at the time of my initial review, but I decided against a direct comparison because the current study implements forward predictions of seismicity rates and magnitude exceedance probabilities, whereas NR2018 focuses solely on history matching. There are a number of history-matching simulation studies that show fluid pressure migration away from injection wells accurately matches earthquake hypocenter locations. As a result, NR2018 revises an existing theme through weak hydromechanical coupling; however, the current study advances the science of injection-induced seismicity by developing a method for forward prediction. In my opinion, the direct comparison between the current study and NR2018 is not a necessary component of the paper (e.g. Fig. 4a, dashed gray line); however, I'll leave it to the authors and editors on whether or not to keep this component of the paper intact. At this point, I highly recommend publication.

Specific Comments: Here are several minor issues that the authors may consider:

Lines 107, 234, & 522: Perhaps change "25000" to "25,000"

Lines 142 – 143: The phrase "...keeps on increasing..." seems a bit informal. Perhaps "...continues increasing..."

With Regards,

Ryan M. Pollyea

Reviewer #3 (Remarks to the Author):

In the revised manuscript the authors provided detailed responses to the issues raised by me and another reviewer in our initial review. The revised manuscript is significantly improved in clarity, readability, and also better conveys the significance of the work. I have no further comments or suggestions and believe the manuscript is now suitable for publication.

Reviewer #1 (Remarks to the Author):

Scope of Review: My expertise is in the field of nonisothermal, multi-phase fluid flow in porous and fractured geologic media with a specific emphasis on modeling subsurface fluid injections. Consequently, the scope of this review is limited to the fluid aspects of fluid-triggered earthquakes. I am not qualified to provide rigorous peer-review in the area of seismological processes.

General Comments: My initial review of the subject manuscript focused on several topics, including the concept of pressure rate (what it means and how it is calculated) and hydrogeological model presentation. In the revised manuscript, the authors adequately address all of my previous concerns, particularly in the context of the pressure rate concept. Specifically, I think the revised Figure 1 is much more accessible to the reader. The additional detail on model parameterization and corresponding rationale also enhances reproducibility and facilitates implementation by third parties.

I have also reviewed the discussion between the authors and Reviewer #3, which focuses on a comparison between the current study and the coupled hydro-mechanical model recently published by Norbeck and Rubenstein (2018). I was aware of this latter study at the time of my initial review, but I decided against a direct comparison because the current study implements forward predictions of seismicity rates and magnitude exceedance probabilities, whereas NR2018 focuses solely on history matching. There are a number of history-matching simulation studies that show fluid pressure migration away from injection wells accurately matches earthquake hypocenter locations. As a result, NR2018 revises an existing theme through weak hydromechanical coupling; however, the current study advances the science of injection-induced seismicity by developing a method for forward prediction. In my opinion, the direct comparison between the current study and NR2018 is not a necessary component of the paper (e.g. Fig. 4a, dashed gray line); however, I'll leave it to the authors and editors on whether or not to keep this component of the paper intact. At this point, I highly recommend publication.

Because Reviewer #3 and the Associate Editor explicitly asked us to show a comparison to the recent study by Norbeck and Rubenstein, 2018, we decided to keep it in the paper.

Specific Comments: Here are several minor issues that the authors may consider:

Lines 107, 234, & 522: Perhaps change "25000" to "25,000"

Done.

Lines 142 – 143: The phrase "...keeps on increasing..." seems a bit informal. Perhaps
"...continues increasing..."

Done.

With Regards,

Ryan M. Pollyea

Reviewer #3 (Remarks to the Author):

In the revised manuscript the authors provided detailed responses to the issues raised by me and another reviewer in our initial review. The revised manuscript is significantly improved in clarity, readability, and also better conveys the significance of the work. I have no further comments or suggestions and believe the manuscript is now suitable for publication.